# Towards heat flux measurements with fast-response fine-wire platinum resistance thermometers on small multicopter uncrewed aerial systems.

Norman Wildmann[1] and Laszlo Györy[1]

[1]Deutsches Zentrum für Luft- und Raumfahrt e.V., Institut für Physik der Atmosphäre, Oberpfaffenhofen, Germany

**Correspondence:** Norman Wildmann (norman.wildmann@dlr.de)

**Abstract.** This study demonstrates the feasibility of measuring temperature variance and heat flux with self-calibrated fine-wire platinum resistance thermometers (FWPRT) on multicopter drones. The sensors are especially designed for light weight, fast response-times and to be carried on miniature drones for turbulence measurements. A significant improvement was found in vertical profiling of temperature gradients compared to slower solid-state sensors, demonstrating reduced hysteresis between ascent and descent phases and accurate representation of strong gradients. More than 100 single flights with the sensors attached to drones of the SWUF-3D fleet were carried out in vicinity to a meteorological mast array at the WiValdi wind energy research park in Northern Germany. The comparison to sonic anemometers shows that temperature variance can be accurately measured within the background flow variability. The same applies for heat flux, which was measured for the first time with multicopter UAS and the eddy covariance method. Heat flux is a crucial parameter to understand the energy balance of the atmospheric boundary layer and turbulent mixing. An uncertainty below 50 W m$^{-2}$ was determined with the constraint that only low wind speed conditions ($<8$ m s$^{-1}$) could be used to allow vertical wind speed measurements with the current algorithm. The results indicate that the temperature sensors are suited for heat flux measurements, but further improvements are necessary with regard to vertical wind speed estimates to decrease the overall uncertainty.

## 1 Introduction

Small uncrewed aerial systems (UAS) are platforms that have been introduced into everyday life within the last decade due to their low cost, good availability, and ease of use. They serve a large variety of applications, including aerial photography and cinematography, but also for scientific Earth observation. In atmospheric sciences, fixed-wing UAS were initially used to collect in-situ measurements, especially in the atmospheric boundary layer (ABL) (van den Kroonenberg et al., 2008; Wildmann et al., 2015). Data collection methods in that case are based on techniques that were established for crewed research aircraft Lenschow et al. (1986). Flow probes and fast-response sensors were installed for the measurement of thermodynamic variables and the derivation of turbulent fluxes Wildmann et al. (2013, 2015). Fixed-wing UAS remain widely used and are becoming standard tools in scientific ABL campaigns (de Boer et al., 2022; Boventer et al., 2024).

This study focuses on small multicopter UAS, which are the most common UAS type and are usually referred to as 'drones'. These systems are easier to operate due to their vertical take-off and landing capability and advanced control systems. In modern

atmospheric measurements, they are most often used to collect vertical profiles of wind, temperature, and humidity (Segales et al., 2020; Lappin et al., 2023). For such profiling tasks, similar sensors as in radiosondes can be deployed and provide good accuracy for temperature and humidity (Hervo et al., 2023). Careful consideration of sensor placement and response time is necessary to optimize data quality (Segales et al., 2022), but high-frequency turbulence data cannot be collected with such sensors. Fuertes et al. (2019) equipped a multirotor with a flow probe and fine-wire thermocouple to allow for turbulence measurements, but did not show results of heat flux estimation. Another alternative is to put sonic anemometers on UAS **?** with the drawback of requiring much larger systems. The smallest resolvable turbulence scales for multicopter UAS depend strongly on rotor size, drone weight and the flow disturbance that is induced, but also on sensor noise, e.g. due to vibrations (Kistner et al., 2024). Wetz and Wildmann (2022) and Wildmann and Wetz (2022) show that with the DLR SWUF-3D (Simultaneous Wind measurement with a UAS Fleet in 3D) quadrotors, turbulence eddies can be resolved with a frequency up to 2 Hz. However, this has been shown only for wind, not for temperature or humidity. High-temporal-resolution sampling of not only wind but also temperature is important for a better understanding of thermodynamic processes in the ABL, especially buoyancy-driven flows (Lappin et al., 2022). This is also crucial in complex terrain, where thermally driven flows often occur and drive exchange processes in valleys (Pfister et al., 2024). A common technique for turbulence flux measurements from stationary measurements in the ABL is the eddy-covariance method (Baldocchi et al., 2001). The eddy-covariance method directly measures the net exchange of gases, heat, and momentum between an ecosystem and the atmosphere by statistically correlating rapid fluctuations in vertical wind speed with concurrent fluctuations in the scalar of interest (e.g., gas concentration or, as in this study, temperature). So far, the eddy-covariance technique to measure heat fluxes has not been directly tried with multicopter UAS without sonic anemometers, because no systems are available that can measure both, vertical wind speed and temperature at a sufficiently high temporal resolution. Lee et al. (2017) calculated heat fluxes with multicopter UAS, but based on a conditional sampling technique that uses mean surface temperature and mean measured temperature at flight height. Further, Greene et al. (2022) use a gradient-based approach to derive fluxes with good results in the arctic stable boundary layer. Typically, sonic anemometers are used to directly measure buoyancy flux from sonic temperature and vertical velocity which is sampled at 10 Hz or higher to resolve a wide range of scales of atmospheric turbulence. The high-resolution parts are particularly important for atmospheric conditions with small integral length scales of turbulence such as in a stable ABL. In order to derive accurate fluxes of sensible and latent heat in the ABL, corrections are necessary which are described in detail in Baldocchi et al. (2001) and Mauder and Foken (2011). In this study, we focus on a comparison of raw measurements between collocated sonic anemometers as a reference and the SWUF-3D UAS. Billesbach et al. (2024) provide a good overview of different methods to derive turbulent fluxes in the ABL and use the EC method (based on sonic anemometer and gas analyzer measurements) as the reference because it is the only method to make direct and independent measurements of the fluxes. It is thus desirable to achieve this with UAS as well. The system presented in Fuertes et al. (2019) has the potential to derive fluxes with the eddy-covariance method on UAS with a fast-response pressure probe and a thermocouple, but they did not do a systematic analysis on fluxes in their study.

The motivation to measuring atmospheric turbulence with a fleet of UAS is to allow obtaining in-situ observations of turbulence and fluxes at flexible locations where it has not been easily possible before, e.g. in very complex terrain, in heights above typical

tower heights or very close to wind turbines. Such measurements allow to quantify the complexity and spatial heterogeneity of boundary-layer processes and potentially derive spatially representative fluxes directly if multiple systems are deployed.

In this study, we present temperature measurements obtained using a platinum fine-wire resistance thermometer (FWPRT), a technique originally applied with fixed-wing UAS (Wildmann et al., 2013). The study wants to answer the following questions:

1. Can FWPRT be placed on a small multicopter UAS so that the measurements represent atmospheric flow and are not significantly disturbed by rotor-induced flow?

2. What is the temporal resolution of the setup and thus the scale of turbulence eddies that can be measured?

3. Do fast temperature measurements in combination with 3D wind estimates from the UAS allow heat flux measurement?

4. What are the uncertainties that can be achieved?

Section 2 describes the sensor design and integration into the UAS, Sect. 3 describes the experimental setup; Sect. 4 gives a description of the applied methods and Sect. 5 shows results of the sensor validation.

## 2 UAS and sensor design and setup

### 2.1 The SWUF-3D UAS

The SWUF-3D UAS are commercially available racing drone frames of type Holybro QAV250. They are powered by a Pixhawk 4 Mini autopilot. Depending on the batteries which are used for the specific operation, the QAV250 can reach flight times up to 25 minutes. In this study, only batteries with a lower capacitiy were available, so that maximum flight times were 15 minutes and therefore, the hover periods were set to a maximum of 12 minutes. Further characteristics of the UAS are described in Wetz et al. (2021). In a fleet configuration, a multitude of drones can fly pre-defined routes synchronously and automatically. Up to twenty drones were operated during the FESSTVaL campaign (Hohenegger et al., 2023). At the research wind park WiValdi, ten drones were operated simultaneously in multiple campaigns before (Wildmann and Kistner, 2024, 2025). Through field tests (Wetz et al., 2021) and wind tunnel calibration (Kistner et al., 2024), the accuracy of wind speed measurement was found to be well below 0.5 m s$^{-1}$ and mostly below 0.3 m s$^{-1}$. The fleet of drones was deployed in the past to investigate spatial correlation and coherence in the ABL (Wetz et al., 2023) as well as wind speed deficit, turbulence and distinct vortices in wind turbine wakes (Wetz and Wildmann, 2023; Wildmann and Kistner, 2024, 2025).

### 2.2 The SWUF-T sensor

The implementation of a compact temperature and humidity sensor on a custom UAS requires a design of a sensor carrier that matches the UAS design. We present here what we call the SWUF-T temperature and humidity module. The concept for the SWUF-T module is to provide a robust commercial and factory-calibrated temperature and humidity sensor along-side a fast and self-calibrated fine-wire platinum resistance thermometer (FWPRT). The commercial sensor is of type HYT.R411 (in

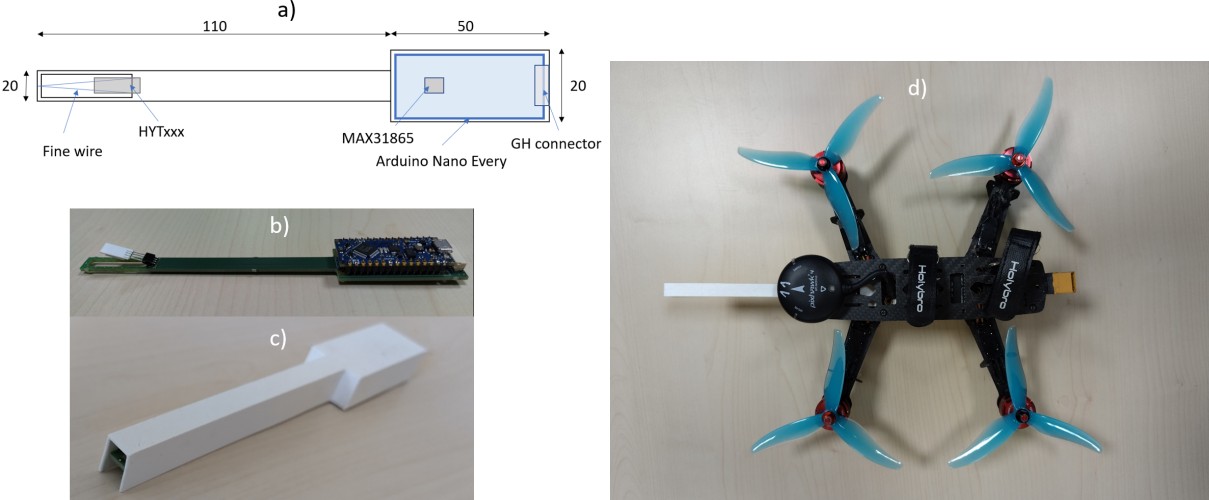

**Figure 1.** SWUF-T board design concept (a) and pictures of the sensor without (b) and with (c) housing. (d) shows the sensor attached to the QAV250 UAS frame.

previous versions HYT271) by Innovative Sensor Technologies (IST AG). The FWPRT consists of a 13 $\mu$m platinum wire of

90 approximately 60 mm length, yielding a resistance $R \approx 50~\Omega$. It is connected to an Analog Devices MAX31865 resistance-to-digital converter chip in a four-wire sensor connection setup. The MAX31865 produces a constant current through a reference resistor (here: 100 $\Omega$) and the platinum wire. The voltage difference is a measure for the resistance of the wire and can be directly converted to a temperature value based on the known temperature-dependent properties of platinum. The I2C interface of the HYT sensors as well as the SPI interface of the MAX31865 are made available at the GH connector of the SWUF-T

board, but can also be routed to an Arduino Nano Every Board that can be attached on top. If the Arduino Nano Every is attached, the GH connector provides a UART interface to the Arduino through which both, HYT and FWPRT measurements are transmitted e.g. via MAVLink protocol. The latter is the setup which is used in this study to connect SWUF-T measurements to the Pixhawk 4 Mini autopilot board. Figure 1 shows the principle design (a) and pictures of the board without (b) and with (c) sensor housing. The housing is designed to shield the sensors from incoming radiation, while allowing passive ventilation.

Wildmann et al. (2013) showed that already slow airspeeds provide sufficient ventilation for forced convection on the thin wire and thus omit radiation and self-heating effects. The radiation can however strongly effect the HYT sensor, so that a shield is recommended.

Ghirardelli et al. (2023) and Jin et al. (2023) recently showed by means of CFD simulations and field measurements that the best position to place a flow sensor on a multicopter is upstream in front of the rotors. Especially since we are operating with

105 the weather-vane mode that will always point the y-axis of the UAS body-frame into the main wind direction, we place the SWUF-T sensor upstream in the main axis, too. The sensing elements of the sensor in our setup is 50 mm in front of the rotor plane. Recent CFD simulations of the QAV250 airframe (Hofmann, 2025) showed that at this position, without any inflow wind speed, a flow of approximately 2 m s$^{-1}$ is induced by the rotors, which means that the sensor will always be ventilated.

As soon as an inflow wind speed is present, the downwash of the rotors is pushed back and the sensor is naturally ventilated. Experiments show that relatively small airflow (i.e. $<2$ m s$^{-1}$) is sufficient to prevent air being trapped in the housing (see App. D).

## 3 Experimental setup

### 3.1 Krummendeich, research wind farm WiValdi

The research wind farm WiValdi at Krummendeich (https://windenergy-researchfarm.com/) in Germany is a facility that is operated by the German Aerospace Center and was instrumented in cooperation with the ForWind universites at Oldenburg, Bremen and Hannover. The meteorological mast array (MMA), which is a set of two 100 m and one 150 m mast in the center, is a suitable infrastructure to validate spatial measurements in the field with UAS and vice versa (i.e. to determine flow disturbance by the masts). Details about the masts are given in Sect. 4.1.

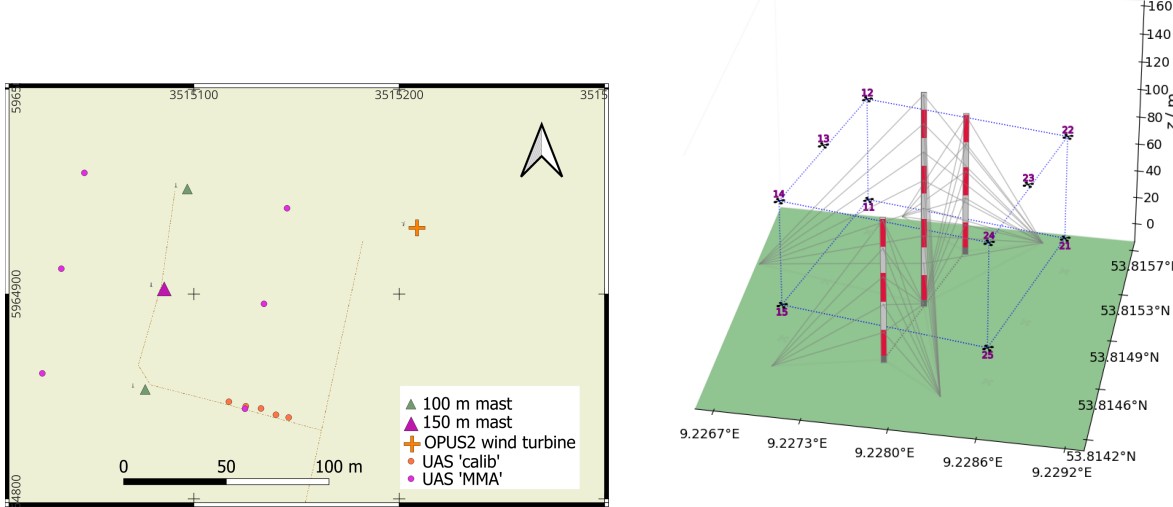

**Figure 2.** Map of the Krummendeich test site with wind turbine OPUS2 (orange), MMA (triangles) and drone flight positions (red circles). Background map: ©OpenStreetMap contributors 2025. Distributed under the Open Data Commons Open Database License (ODbL) v1.0. On the right, the MMA-pattern is shown in a 3D representation.

The analyses in this study are based on a campaign that was conducted in July 2024, particularly from 22-25 July 2024, when flights were conducted in close proximity to the mast array. During these days, the wind turbines OPUS1 and OPUS2 at the site were not operating. Figure 2 shows a map with hover positions of the UAS relative to the masts and Tab. 1 gives a list of flights that were used for the analyses with the corresponding meteorological conditions. Two different flight pattern ´MMA'

**Table 1.** Measurement flights from 22-25 July 2024

| # | $t$ | pattern | $z$ | $\Psi$ | $U$ | $TI$ | $T$ | $\varphi$ | clouds |
|---|---|---|---|---|---|---|---|---|---|
| | UTC | | m | deg | m s$^{-1}$ | % | °C | % | 1/8 |
| 36 | 22.07.2024 10:10 | MMA | 25, 90 | 303 | 8.4 | 19 | 16.7 | 67 | 8/8 |
| 37 | 22.07.2024 10:50 | MMA | 25, 90 | 300 | 8.2 | 18.4 | 16.8 | 66 | 7/8 |
| 38 | 22.07.2024 11:30 | MMA | 25, 90 | 303 | 8.0 | 18.7 | 16.9 | 71 | 6/8 |
| 39 | 22.07.2024 12:30 | MMA | 25, 90 | 303 | 7.9 | 20.4 | 17.1 | 74 | 4/8 |
| 40 | 22.07.2024 12:55 | MMA | 25, 90 | 300 | 8.1 | 20.6 | 17.2 | 73 | 4/8 |
| 41 | 22.07.2024 13:30 | MMA | 25, 90 | 292 | 8.5 | 18.8 | 17.3 | 69 | 4/8 |
| 42 | 22.07.2024 14:15 | MMA | 25, 90 | 309 | 7.8 | 17.4 | 17.3 | 70 | 4/8 |
| 43 | 22.07.2024 14:50 | MMA | 25, 90 | 312 | 7.5 | 18.6 | 17.4 | 68 | 2/8 |
| 44 | 22.07.2024 15:15 | MMA | 25, 90 | 305 | 6.6 | 19.8 | 17.4 | 68 | 1/8 |
| 46 | 23.07.2024 06:40 | MMA | 25, 90 | 228 | 6.6 | 15.7 | 19.0 | 82 | 7/8 |
| 47 | 23.07.2024 07:20 | MMA | 25, 90 | 246 | 5.8 | 14.5 | 19.2 | 81 | 7/8 |
| 48 | 23.07.2024 07:55 | MMA | 25, 90 | 256 | 5.1 | 21.4 | 19.4 | 80 | 7/8 |
| 50 | 23.07.2024 08:40 | MMA | 25, 90 | 247 | 6.2 | 16.5 | 19.7 | 78 | 7/8 |
| 51 | 23.07.2024 09:20 | MMA | 25, 90 | 243 | 5.4 | 18.4 | 19.7 | 77 | 8/8 |
| 52 | 23.07.2024 09:45 | MMA | 25, 90 | 238 | 7.1 | 14.0 | 19.9 | 75 | 8/8 |
| 69 | 25.07.2024 07:45 | calib | 99 | 224 | 3.9 | 19.8 | 15.3 | 75 | 7/8 |
| 70 | 25.07.2024 08:15 | calib | 99 | 222 | 4.0 | 22.2 | 15.8 | 72 | 7/8 |
| 74 | 25.07.2024 11:55 | MMA | 25, 90 | 212 | 5.0 | 15.1 | 19.8 | 48 | 7/8 |
| 77 | 25.07.2024 13:00 | MMA | 25, 90 | 204 | 2.9 | 27.9 | 20.5 | 47 | 7/8 |
| 78 | 25.07.2024 13:30 | MMA | 25, 90 | 214.5 | 3.6 | 21.8 | 21.2 | 41 | 7/8 |

and 'calib´ are listed and correspond to the pink and orange dots of UAS positions in Fig. 2 respectively. The ´MMA´ pattern was designed to mimic sensor positions of the mast array in a further plane perpendicular to the mast array axis, whereas the 'calib´ pattern was used to have five UAS as close as possible to each other for cross-calibration of wind and temperature measurements. A comprehensive table of which drones were flown at which position during each flight is provided in App. A.

## 3.2 Cochstedt, National Test Centre for Unmanned Aircraft Systems

The National Test Centre for Unmanned Aircraft Systems at the airport of Cochstedt is a facility operated by the German Aerospace Center, specifically designed to support UAS operations with its infrastructure. An airport traffic zone (ATZ) can be activated up to 3000 feet above ground level to mitigate air risk for UAS operation. We used this possibility to conduct vertical profiles with our SWUF-3D drones in June 2024. Most of the flights were conducted using a newer version and frame of a multicopter; however, we also conducted one flight on June 25, 2024, up to 300 meters in height with the frame as described in Sect. 2 which is used in this study.

**Table 2.** List of 3d sonic anemometers at the MMA that were used in this study and the applied temperature offset.

| ID | tower | height | boom direction | $\Delta T_s$ |
|---|---|---|---|---|
| 500706 | North | 25 m | South | -1 K |
| 500690 | North | 99 m | Top | 0 |
| 500772 | South | 25 m | North | -1.2 K |
| 500753 | South | 99 m | Top | -0.7 K |
| 500727 | Center | 100 m | South | 0 K |

## 4 Methods

### 4.1 Meteorological mast array

The meteorological mast array (MMA), conceptualized by the University of Oldenburg, features 51 3D sonic anemometers and 32 cup anemometers to measure distributed wind fields within an area covering the rotor diameter of the installed wind turbine at the WiValdi site. The three masts are separated by 50 m, with the central mast reaching 150 m and the outer masts reaching 100 m. In this study, we use five sonic anemometers as a reference for the drone-based measurements. The position and identification of these sonic anemometers are listed in Table 2.

The sonic anemometers at 25 m height are installed on 7 m booms pointing towards the central mast; those at 99 m are installed on top of the masts to reach their final height. Each Thies Clima 3D sonic anemometer provides three-dimensional wind information in the anemometer's coordinate system and three readings of sonic temperature (see Sect. 4.2). The raw data of the instruments is available at a frequency of 10 Hz. A cross-wind correction is performed internally by the sonic anemometer according to Liu et al. (2001) and we use the median of the three measured temperatures ($T_u$, $T_v$ and $T_w$) to obtain a final temperature reading, $T_s$, as suggested by the manufacturer:

$$T_s = \mathrm{median}(T_u, T_v, T_w) \tag{1}$$

For turbulence and heat flux measurements, accurate absolute temperature is not critical. The raw temperature data from the sonic anemometers is not well calibrated and exhibits offsets compared to the drone measurements and the well-calibrated temperature sensor at the WiValdi site's inflow mast, located approximately 600 m to the west (out of scale in Fig. 2). For the three days of flights analyzed in this study, we apply a rough bias correction based on the inflow mast measurements. Details of all the used sensors at the WiValdi masts is given in App. B. For spatial temperature measurements with the MMA, improved corrections will be necessary in future work if high absolute accuracy is required.

### 4.2 Sonic temperature

The in-flight performance of the temperature sensor requires validation through field experiments. The preferred method is to fly the UAS in close proximity to sonic anemometers. A sonic anemometer provides a temperature reading derived from the

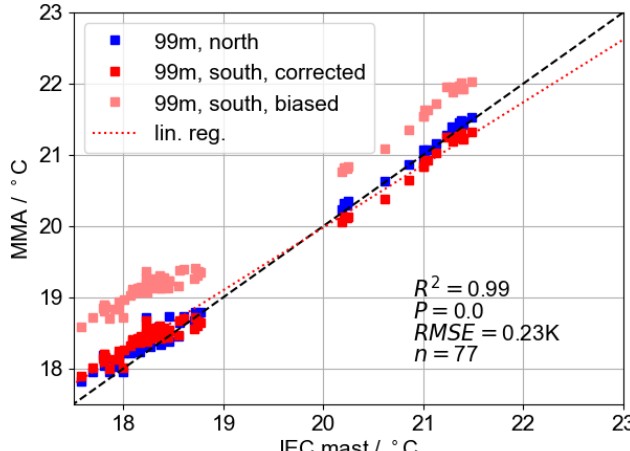

**Figure 3.** 10-minute mean measurements of temperature of North and South mast sonic anemometer at 99 m compared to a reference temperature sensor at the IEC inflow mast.

speed of sound, which is often referred to the 'sonic temperature', $T_s$. This value is close to, but not equal to virtual temperature $T_v$. A common equation to calculate sonic temperature from air temperature $T$ is according to Schotanus et al. (1983):

$$T_s = (T + 273.15)(1.0 + 0.51q) \quad , \tag{2}$$

where $q$ is the specific humidity. The SWUF-T sensor also measures humidity, so that measured air temperature by the FWPRT temperature sensor can approximately be converted to an equivalent sonic temperature more easily than vice versa. Therefore, all temperature readings from the UAS that are compared with sonic anemometer measurements are converted to sonic temperature according to Eq. 2.

### 4.3 Heat flux

Using the eddy covariance method, sensible heat flux, H, can be calculated from variations in temperature and vertical wind speed as

$$H = \rho C_p \overline{T'w'} \tag{3}$$

where $\rho$ is the air density, $C_p$ is the specific heat capacity of air and $\overline{T'w'}$ the covariance of temperature and vertical wind speed. Since the sonic anemometers do not directly measure $\overline{T'w'}$, but the buoyancy flux $\overline{T_s'w'}$ (Liu et al., 2001), we use this

parameter throughout this study for the comparison between UAS and sonic:

$$H_b = \rho C_p \overline{T_s'w'} \tag{4}$$

Both variables $\rho$ and $C_p$ can be calculated from thermodynamic measurements made by the drone itself, namely temperature, humidity, and pressure. We apply a correction for moisture to the specific heat capacity of dry air, such that

$$C_p = 1004.67 \text{ J kg}^{-1}\text{K}^{-1} \left(1 + 0.84 m_h\right) \tag{5}$$

where $m_h$ is the water vapor mixing ratio.

We intentionally calculate buoyancy flux using 10-minute averages, recognizing that a representative flux may require longer averaging periods to adequately sample the largest turbulent eddies for the eddy covariance method. However, the purpose of this paper is not to study heat flux itself within the atmospheric boundary layer, but rather to determine the capability of drones equipped with fast temperature sensors to derive heat flux values comparable to those from other in situ sensors, especially

in the small scales. At the current stage of development in this study, the SWUF-3D fleet was unable to fly for 30 minutes or longer. The most consistent, extensive and comparable dataset was obtained using 10-minute averages.

## 4.4    Laboratory calibration

The HYT sensors come factory-calibrated with a specified accuracy of 0.2 K in the temperature range $0 \ldots 50°\text{C}$. The MAX31865 defines a total accuracy of temperature measurements over all operating conditions of 0.5 K. In our setup, we do not use a stan-

dard PT100 or PT1000 element, but use a platinum fine wire as specified in Sect. 2. From a known base resistance of the wire at $0°\text{C}$ ($R_0$) the actual resistance $R_{\mathrm{pt}}$ is a function of temperature $T$ accrding to the Callendar–Van Dusen equation (Callendar, 1887; Van Dusen, 1925) for temperatures above $0°\text{C}$:

$$R_{\mathrm{pt}} = R_0 \left[1 + aT + bT^2\right] \quad , \tag{6}$$

where $a$ and $b$ are material specific coefficients.

The length of the wire varies due to manufacturing inaccuracies, so that $R_0 = 50 \pm 5 \ \Omega$ are possible in the manufacturing process. For best accuracy, we calibrate each sensor individually in an EdgeTech RHCal portable calibration chamber with certified and traceable accuracy of $0.1°\text{C}$. We use the device's full temperature range of $10 \ldots 50°\text{C}$ with a step size of $5°\text{C}$. Figure 4 shows the calibration curve of five sensors compared to the theoretical curve of a platinum resistance with $R_0 = 50 \ \Omega$.

The curves show a strong linear relationship between temperature and resistance in the calibration range, which corresponds to the theoretical expectations. Applying a linear model to the theoretical platinum curve within the considered temperature range results in a coefficient of determination of $R^2 = 1 - 2.86 \times 10^{-6}$, indicating a very low calibration error. Due to the temperature limitations of the EdgeTech RHCal calibration chamber, we cannot measure $R_0$ directly. Instead, we fit the measured resistance values to the known calibration temperatures with a first-order polynomial:

$T_{\mathrm{pt}} = c_1 \cdot R_{\mathrm{pt}} + c_0 \quad . \tag{7}$

We obtain two coefficients $c_0$ and $c_1$ for each FWPRT sensor (see Tab. C1). These coefficients are then used to calculate the temperature $T_{\mathrm{pt}}$ based on the measured resistance $R_{\mathrm{pt}}$ of the platinum fine wire. Applying this linear model to the measurement

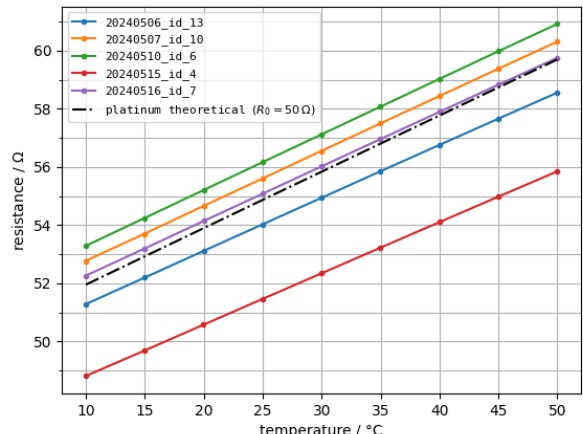

**Figure 4.** Calibration curve of five individual FWPRT sensors and the theoretical curve of a platinum resistance at $R_0 = 50\,\Omega$ (dashed line).

series now produces a median $R^2 = 1 - 6.57 \times 10^{-6}$. The coefficients of determination exceed 0.99, indicating a good linear relationship and confirming the suitability of a linear calibration model for our sensors. Observed deviations are primarily

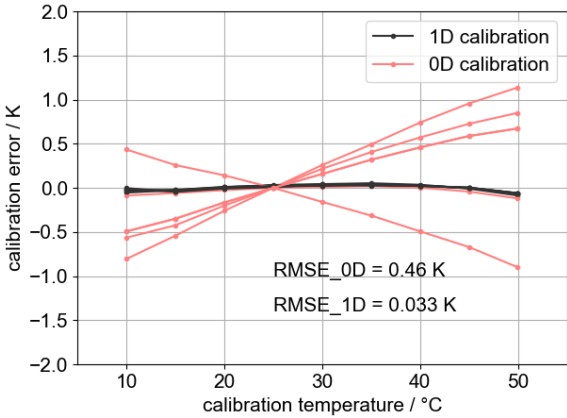

**Figure 5.** Calibration error of 5 FWPRT sensors with respect to the reference sensor in the calibration chamber after 0 D calibration (bias correction at 25°C, red curves) and 1 D calibration (grey curves).

biases, which can be corrected by adjusting resistance values. Fig. 5 shows the errors for the whole calibration range if only a bias correction is applied (red curves) in comparison to the polynomial fit (black curves). While an individual bias correction (0-D calibration) provides a reasonable approximation, it shows that the differences in the slopes of the individual curves are too large to achieve the desired precision of less than 0.1 K over the full temperature range. These variations in slope may be attributed to material impurities and production tolerances. Therefore, a slope adjustment by the previously mentioned first-

order calibration equation with a second coefficient becomes necessary. Not all 25 temperature sensors, including spares, could readily be calibrated in the calibration chamber for the campaign so that some were only calibrated with the bias correction. We thus expect an average absolute RMSE$<= 0.5$ K. The calibration coefficients for the calibrated temperature sensors that were used in this study are listed in App. C in Tab. C1.

### 4.5 Vertical velocity esimate

Attempting to calculate fluxes with the eddy covariance method requires synchronous measurements of temperature with vertical flow velocity $w$. As described in Wildmann and Wetz (2022), vertical velocity can be estimated from thrust and lift of the UAS. The force acting on the drone in $z$-direction $F_z$ in the body frame of the drone is a combination of gravitational force ($mg$ rotated into body frame with roll $\varphi$ and pitch $\theta$ angle), vertical accelerational forces $m\ddot{z}$, thrust $T$ and lift force $F_L$ which depends on the drag force $F_x$:

$$F_z = -mg\cos(\theta)\cos(\varphi) + m\ddot{z} + T + F_L(F_x) \tag{8}$$

Vertical velocity is derived based on a calibrated curve using the equations

$$w_b = \begin{cases} c_{z\uparrow} F_z^{b_z\uparrow} & F_z \geq 0 \\ c_{z\downarrow} F_z^{b_z\downarrow} & F_z < 0 \end{cases}. \tag{9}$$

Since the study in 2022, new rotors and new batteries were installed on the SWUF-3D UAS, which changes the thrust and lift behaviour. Parameters were adjusted accordingly. It also showed that it is beneficial to first rotate the forces into the geodetic coordinate system and do the vertical velocity calibration in this frame of reference, so that:

$$w = \begin{cases} c_{z\uparrow} F_z^{g_z\uparrow} & F_z \geq 0 \\ c_{z\downarrow} F_z^{g_z\downarrow} & F_z < 0 \end{cases}. \tag{10}$$

## 5 Results

### 5.1 Flow disturbance and sensor placement

From Wetz and Wildmann (2022) and Wildmann and Wetz (2022), we know that the resolution of turbulence by measuring the three-dimensional wind vector with the SWUF-3D fleet is limited to approximately 2 Hz, corresponding to a 2.5 m eddy size at a mean wind speed of 5 m s$^{-1}$. To demonstrate that the flow at these scales is not significantly disturbed at the temperature sensor's position, we visualized the flow using smoke and flew the drone within the plume. At wind speeds of 2–3 m s$^{-1}$, recorded during the test, it is clear that the flow is significantly disturbed only by the downwash below and downstream of the UAS. Virtually no disturbance is detected directly in front of the UAS. At higher wind speeds, upstream effects will be

even smaller. Figure 6 shows a snapshot of the flow visualization. Previous studies using CFD (Ghirardelli et al., 2023) and lidar measurements (Jin et al., 2023) have shown the effects of rotor downwash on sensor position for much larger drones but reached the same conclusion: placement in front of the drone, aligned with the wind direction, is the optimal location. Operating the drone in weather vane mode, i.e., turning the nose into the wind, is essential for good measurement accuracy. A small experiment was also setup to show that the sensor housing alone does not trap the flow at flow velocities as low as 1-2 m s$^{-1}$ and is described in App. D.

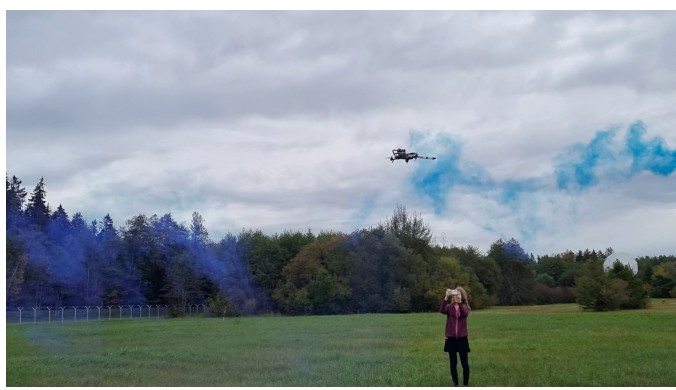

**Figure 6.** Flow visualization with blue smoke.

## 5.2 Average temperature

As typical for such instruments, the sonic anemometers at the met mast array do not provide a precise measurement of the absolute virtual temperature value and were corrected accordingly as described in Sect. 3. To show that the calibrated SWUF-T sensors provide robust temperature readings, we show the comparison of five UAS that were flown in close proximity
(calib_pattern, see Sect. 3) in Fig. 7. In the background, the corrected sonic temperatures at 99 m are shown for the south (solid blue) and center (light blue) mast. It shows that the temperature readings of QAV21 have somewhat smaller variance, which is consistent throughout the experiment and may be due contamination or other issues with the platinum wire that can unfortunately not be traced back.

The average relative deviations between the four sensors on the UAS are below 0.1 K for this test. The deviations to the sonic
anemometers are larger which can be attributed to the distance of 50 m to the mast. It is evident that the fluctuations within the 10-minute period fit better to the south mast anemometer than the center mast which is even further away.

## 5.3 Vertical profiling

Temperature sensors are known to introduce errors in vertical profiling with UAS (see Cassano (2014) for fixed-wing and Tikhomirov et al. (2021) for multicopters), as well as with radiosondes (Tschudin and Schroeder, 2013), when slow sensor
response times meet temperature gradients during ascent (and descent). These errors can be corrected if the sensor's time

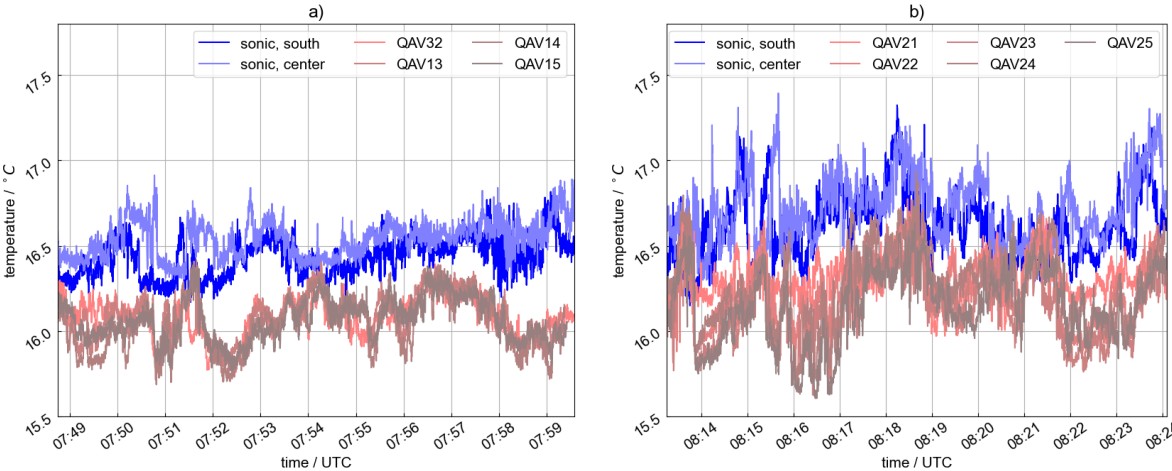

**Figure 7.** Time series of virtual temperature for five UAS (red) in close proximity to the met mast array and corresponding sonic anemometer readings (blue) for flight #69 (a) and #70 (b).

constant is well known (Cassano, 2014), but this requires rigid filtering of high-frequency noise, which can also eliminate information about small-scale turbulence. A fast sensor is always preferable for accurate soundings. Unlike radiosonde sensors, our sensor is not exposed to very low temperatures and pressures, which are particularly unfavorable conditions. The SWUF-3D fleet is designed for boundary-layer operation in non-precipitation conditions and can therefore be optimized for these
conditions. Figure 8 shows two examples of vertical profiles measured simultaneously with an HYT sensor and a FWPRT. The hysteresis caused by the HYT sensor's time response is clearly evident in Fig. 8a) and results in a temperature offset of almost 1°C between ascent and descent. No hysteresis is observed with the FWPRT sensor (red curves). The FWPRT data does indicate that more turbulent fluctuations are observed during ascent, likely due to mixing by rotor downwash during descent. The dotted blue lines show a time-lag-corrected version of the HYT measurements, which shows good agreement in this case
with a constant temperature gradient. For the time-lag-correction, a time constant of 20 s is used and data is filtered with a 5 s moving average to avoid noise amplification before. Figure 8b) provides an example of a more complex boundary layer in the early morning, where a nighttime inversion is still present but is being slowly eroded by surface heating below 100 m. Between ascent and descent, a 10-minute hover period is included in this flight during which the temperature at 150 m significantly cools from the bottom. This cooling can be seen in the measurements by the fast sensor, but not by the slow sensor, particularly
because the temperature increase during ascent was not accurately measured. Additionally, the vertical profile was interrupted at a height of 100 m for 10 s during ascent and descent, which is clearly reflected in a temperature adjustment for the slow sensor (dark blue line) due to its response time. Time response correction brings the slow sensor's curves significantly closer to the fast sensor's curves but is highly sensitive to the applied time constant and can therefore introduce large errors. Both examples demonstrate the importance of fast temperature sensors for profiling the atmospheric boundary layer with the goal of
resolving small-scale structures, accurate inversions, and temperature gradients.

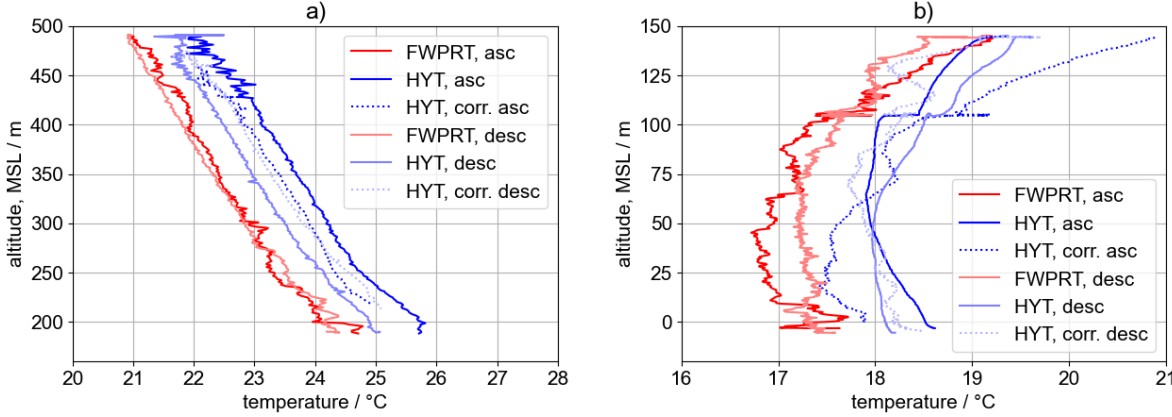

**Figure 8.** Vertical profile at Cochstedt airport. Comparison between SWUF-T fine-wire sensor (red) and HYT temperature sensor with (blue, dotted) and without (blue, solid) time constant correction. a) shows a vertical profile up to 300 m at the Cochstedt airport on 25 June 2024, 15:00 UTC, b) shows a vertical profile up to 150 m at Krummendeich in the early morning of 20 July 2024, 04:40 UTC.

## 5.4 Turbulence measurements

We developed and integrated the SWUF-T module into the SWUF-3D fleet with the primary goal of resolving small-scale turbulent temperature fluctuations and determining second-order statistics with accuracy comparable to that of other in situ instrumentation. This chapter presents the results of in-flight tests of the SWUF-T module, comparing its measurements with those from sonic anemometers at the MMA in the WiValdi research park.

### 5.4.1 Calibration and inter-drone comparison

The calibration flight pattern with five simultaneously hovering drones spaced 8 m apart (see Sect. 5.2) is well suited for inter-comparison of turbulence measurements from the ten drones used in the fleet. We calculate mean temperature ($\overline{T}$), temperature variance ($\sigma_T^2$), vertical wind speed variance ($\sigma_w^2$), and heat flux ($H$) for the five drones during flights #69 and #70. QAV12 exhibited some temperature reading dropouts during these flights, which particularly affected the variance estimation; therefore, its data is grayed out. The temperature sensor of QAV21 consistently showed low variance values throughout the campaign, a problem identified through this test and is likely due to problems with the platinum wire. While QAV12 can still be used in flights without temperature sensor dropouts, QAV21 is excluded from further analysis in this study. The agreement of variance and heat flux estimates from the drones in flight #70 is very good within the fleet and compared to the sonic anemometer ($\Delta H < 10$ W m$^{-2}$). During flight #69, lower turbulence was observed; consequently, the drones show an overestimation of variances and heat flux ($\overline{\Delta H} = 40$ W m$^{-2}$) for this flight. We attribute this to the physical limitations of turbulence measurement at small scales with these drones in their current state of development. Table 3 presents all derived values for the calibration flights.

**Table 3.** Calibration flight results.

| Flight #69 | | | | | | |
|---|---|---|---|---|---|---|
| Variable | Sonic | QAV32 | QAV12 | QAV13 | QAV14 | QAV15 |
| $\overline{T}$ / °C | 16.35 | 16.3 | 16.4 | 16.3 | 16.3 | 16.3 |
| $\sigma_T^2$ / K$^2$ | 0.013 | 0.017 | 0.0028 | 0.021 | 0.03 | 0.019 |
| $\sigma_w^2$ / m$^2$s$^{-2}$ | 0.3 | 0.47 | 0.1 | 0.43 | 0.49 | 0.62 |
| $H$ / W m$^{-2}$ | 35 | 59 | NA | 70 | 98 | 73 |
| Flight #70 | | | | | | |
| Variable | Sonic | QAV21 | QAV22 | QAV23 | QAV24 | QAV25 |
| $\overline{T}$ / °C | 16.55 | 16.5 | 16.4 | 16.3 | 16.5 | 16.3 |
| $\sigma_T^2$ / K$^2$ | 0.044 | 0.019 | 0.041 | 0.041 | 0.058 | 0.054 |
| $\sigma_w^2$ / m$^2$s$^{-2}$ | 0.76 | 0.58 | 0.65 | 0.67 | 0.76 | 0.48 |
| $H$ / W m$^{-2}$ | 124 | 12 | 126 | 133 | 133 | 129 |

Figure 9 shows the variance spectra for flight #70. The spectra of the four SWUF-T sensors (red) follow the Kolmogorov
$-5/3$-slope and the sonic anemometer reference well up to the frequency of 2 Hz, which is just below half the sampling
frequency that was set to 5 Hz for the experiment. We also show the slow HYT-sensor in comparison (green) which are unable
to resolve turbulence scales appropriately above $10^{-2}$ Hz. This illustrates the huge improvement with the new sensors.

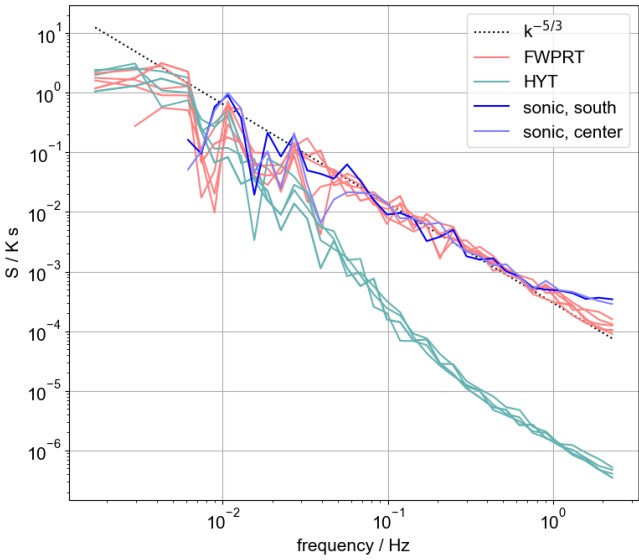

**Figure 9.** Spectra of sonic temperature for UAS flight #70 next to a sonic anemometer showing measurements of the sonic (blue), the slow
HYT271 (green) sensor and the FWPRT (red).

### 5.4.2 Comparison to MMA

For all flights listed in Table 1, we determined the temperature variance during the drone hover periods and the corresponding
time periods and height levels of the sonic anemometers at the mast array.

Figure 10b shows a scatter plot of this comparison, separating measurements at 25 m (blue) and 99 m (red) above ground
level. A coefficient of determination $R^2 = 0.9$ is achieved. At higher variance values, the scatter increases slightly. This may
be attributable to the increasing effect of spatial separation between the UAS and mast under conditions of higher turbulence.
To provide an indication of the achievable agreement under these conditions, we also present a comparison between the two
sonic anemometers at the north and south masts of the MMA at the two levels. The resulting coefficient of determination is
very high at both levels ($R^2_{25} = 0.98$ and $R^2_{99} = 0.99$, but RMSE is slightly lower than for the UAS comparison ($RMSE_{25} =$
$RMSE_{99} = 0.04\ K^2$ vs. $RMSE = 0.02\ K^2$). This demonstrates that the uncertainty of the experimental setup exceeds the
uncertainty of the sensors themselves and therefore does not represent the actual limit of accuracy; however, it does show that
the sensor uncertainty is equal to or smaller than the experimental uncertainty.

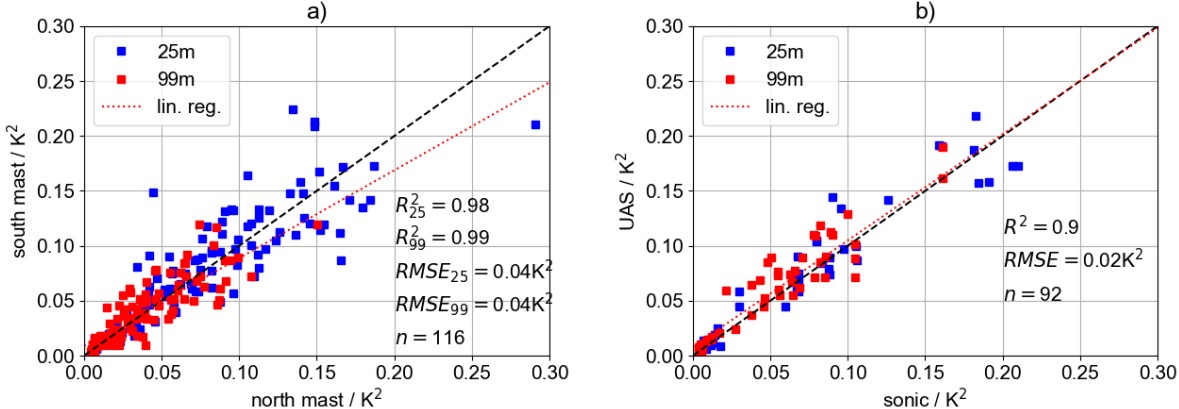

**Figure 10.** Comparison of temperature variance between sonic anemometers of north and south mast (a) and between UAS and the closest
mast (b). Shown are the comparisons at 25 m level (blue) and 99 m level (red). The linear regression is shown as red dotted line.

### 5.4.3 Heat flux

With fast temperature measurements and simultaneous vertical wind measurements, as demonstrated in Wildmann and Wetz
(2022), heat flux calculations are possible if the dominant scales for turbulent transport are large enough to be sampled at a
2 Hz temporal resolution (i.e., approximately 2...5 m eddy size at 4...10 m s$^{-1}$).

Several boundary conditions must be considered when comparing heat flux measurements from the SWUF-3D UAS and
sonic anemometers at the MMA:

1. Only conditions with horizontal wind speeds below 8 m s$^{-1}$ should be considered because vertical wind speed measurements from the drones are unreliable at higher wind speeds with the current vertical wind estimation algorithm Wildmann and Wetz (2022).

2. Heat flux in the ABL can be highly variable in time and space, and the 10-minute flight periods do not necessarily represent stationary conditions but may include transient flows. On the other hand, ten minutes can also be too short to get a good statistical representation of large eddies.

3. No study has yet examined the effect of the masts and guy wires on turbulence measurements at the MMA; therefore, the sonic anemometer reference may also contain errors. The UAS flights in this study were conducted with wind directions from 200–300°, minimizing the likelihood of major disturbances.

To establish a benchmark for the expected uncertainty in a comparative experiment, we present a comparison between sonic anemometers at the MMA - specifically, the ones at 99 m on the south and north mast and the ones at 25 m for the both masts (see Fig. 11a). Although these two masts are only 100 m apart, the coefficient of determination between heat flux estimates is only $R^2 \approx 0.71$ at the 25-m level and $R^2 \approx 0.78$ at the 99-m level. we show the numbers individually for the two heights to show that there is no large difference in the experiment, especially in the RMSE. For the comparison with the UAS, we are interested in the overall accuracy and thus give the combined values. Figure 11b shows the comparison of each available UAS flight meeting the filter criteria with the closest sonic anemometer (typically at a distance of $d \approx 50$ m). The correlation is higher than that between the masts ($R^2 = 0.84$). The RMSE is 38 W m$^{-2}$. As with the variance comparison, this indicates that the accuracy of the drone measurements falls within the uncertainty of the experimental setup. As in the calibration flight example, very low heat fluxes are overestimated by the drones, and scatter increases with higher $H$ values. In the figure, all calculated values are shown in light colors, while dark/full colors represent only those values remaining after filtering for wind speeds exceeding 8 m s$^{-1}$. This shows that vertical wind speed variance is a significant source of uncertainty for flux measurements with the SWUF-3D fleet. The scatter plots for vertical velocity variance for mast intercomparison and UAS to mast comparison are given in App. E.

Figure 12 shows the time series of measured and derived variables for 22 July 2024, the day with the most consecutive flights between 10:00 UTC and 16:00 UTC. On the left, drones at 25 m hover height are compared to the corresponding sonic anemometers at the same height. For this height and day, mainly two drones, i.e. QAV15 (light red circles) and QAV25 (dark red stars), were measuring without technical problems of the temperature sensors, while QAV21 and QAV32 at the northern mast did have failures. QAV15 is placed west of the mast and QAV25 on the east. The plot reveals several interesting observations:

– The SWUF-T sensors accurately capture the long-term temperature trend (Fig. 12a&e).

– The temporal variation of temperature variance at 25 m is substantial (Fig. 12b), meaning that small offsets in time and space can lead to significant errors in direct comparisons between sonic anemometer and drone measurements. The same applies to vertical wind speed variance at 99 m (Fig. 12f).

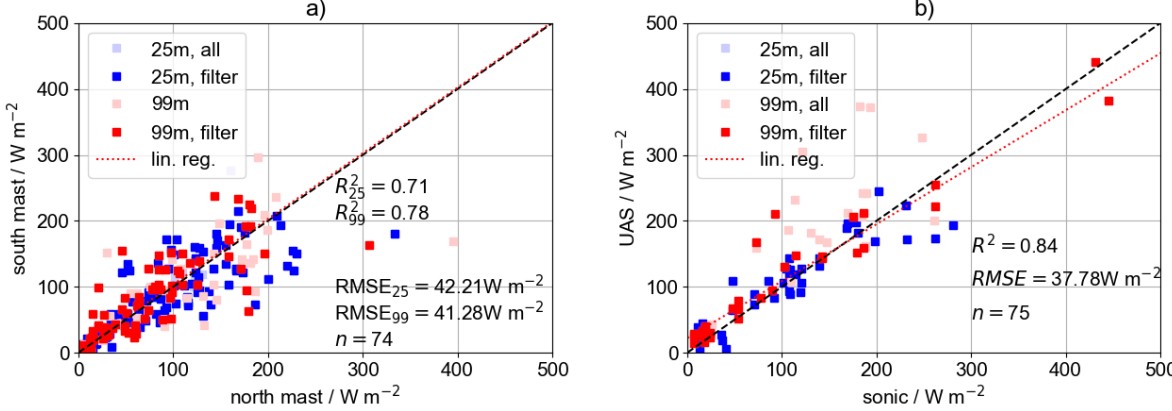

**Figure 11.** Comparison of heat flux between sonic anemometers of north and south mast (a) and between UAS and the closest mast (b). Shown are the comparisons at 25 m level (blue) and 99 m level (red). The linear regression is shown as red dotted line.

– The observed buoyancy flux is variable and features some distinct peaks, e.g. at 12:30 UTC at 25 m (Fig. 12d). Even more variability is found at 99 m (Fig. 12h). Some measurements by the drones are significantly higher than the sonic mast measurements, but within the range of values that is observed by the sonics during the whole period.

## 6 Conclusions

This study has demonstrated the feasibility of robust temperature measurements using self-calibrated fine-wire resistance thermometers (FWPRTs) integrated into the SWUF-3D UAS fleet. Key findings and conclusions are summarized below:

Turbulent temperature fluctuations at scales larger than 1 m can be effectively measured using the SWUF-3D fleet equipped with SWUF-T sensor modules. The physical limits are probably not even reached, since platinum wires with the used diameter have been shown to resolve scales up to 10 Hz (Wildmann et al., 2013), but sampling rate was technically limited to 5 Hz for the SWUF-T sensors at this point. We conclude that rotor-induced flow disturbance does not appear to significantly impact temperature measurements, even with the sensor positioned close to the rotor plane, provided it remains aligned with the wind direction. Compared to slower solid-state sensors, the SWUF-T FWPRT represent a substantial improvement for vertical temperature gradient profiling in the ABL. The absence of hysteresis between ascent and descent allows both flight phases to be used for future investigations of mean boundary-layer development.

While calculating turbulent fluxes using the eddy covariance method is possible, it is susceptible to errors in both temperature and vertical wind speed measurements. The experimental setup, characterized by high wind speeds and turbulence intensity, presented challenges in precisely quantifying the uncertainty of flux measurements with the SWUF-3D fleet. The spatial variability of heat flux, calculated from 10-minute averages of sonic anemometers, was found to be on the order of 42 W m$^{-2}$, comparable to the observed RMSE for the drone measurements (38 W m$^{-2}$).

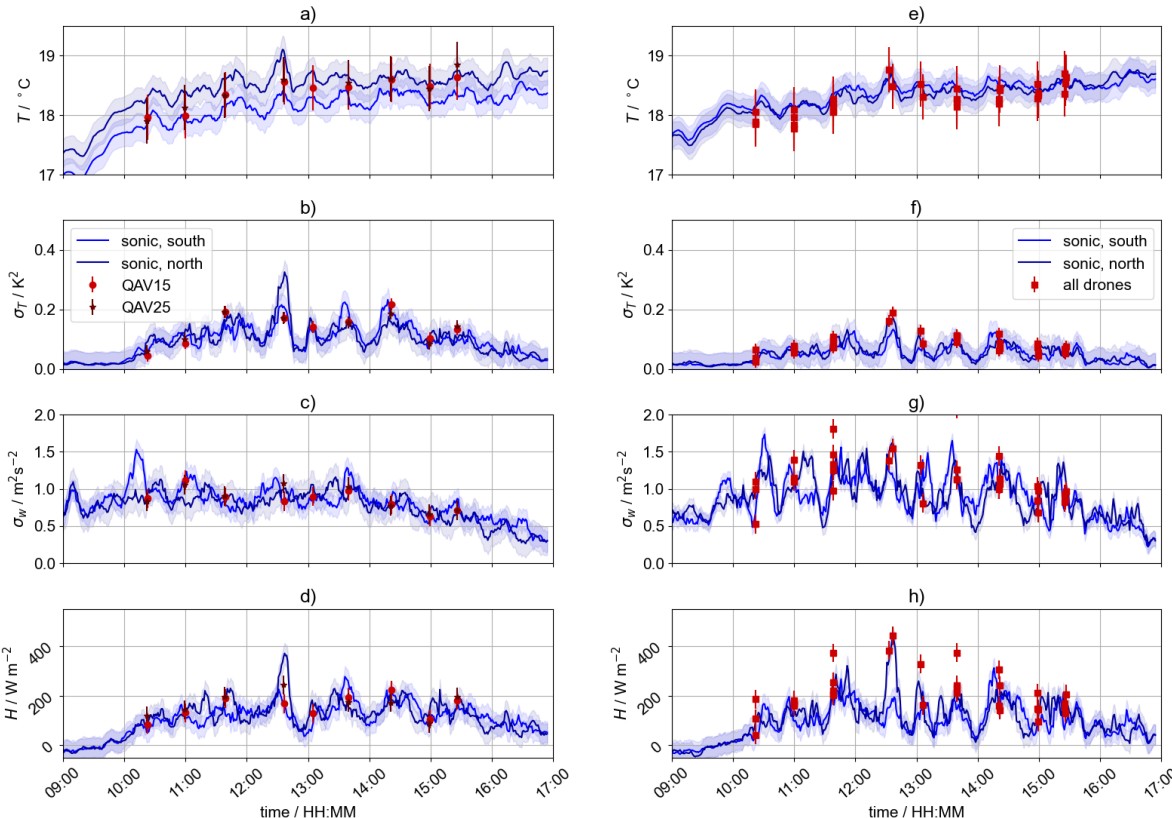

**Figure 12.** Time series of temperature $T$ (a,e), temperature variance $\sigma_T^2$ (b,f), vertical wind speed variance $\sigma_w^2$ (c,g) and heat flux $H$ (d,h) for 22 July 2024. 10-minute averaged sonic anemometer measurements on south (dark blue) and north mast (light blue) are compared to drone measurements (red squares).

Future work should focus on extending flight times (beyond 30 minutes) and conducting measurements under more stationary and homogeneous flow conditions and a wider range of atmospheric conditions (e.g. stable ABL, weak turbulence, free convection) to enhance confidence in the results and refine uncertainty estimates. The analysis revealed that uncertainties in heat flux are to a large degree related to uncertainties in vertical wind speed estimation, particularly at high wind speeds, where vertical flux estimates cannot currently be considered reliable. Further improvements to vertical wind speed measurement are therefore crucial for reducing flux measurement uncertainty. It is also evident, as has already been reported in Wetz and Wildmann (2022) and Wildmann and Wetz (2022) that very small-scale turbulence, as in stable boundary layers, cannot be sampled at the current state of development. It is necessary to reduce vibration noise and frame and rotor size to achieve smaller scales.

Comparisons with the meteorological mast array (MMA) proved challenging due to uncertainties in the sonic anemometers' temperature and wind measurements under varying inflow conditions. Further investigation of flow around the MMA should be performed, for which the SWUF-3D fleet itself presents a valuable tool.

Despite these challenges, the flexibility and robustness of the SWUF-3D fleet and UAS in general are significant assets. Their deployment in future field campaigns, particularly in locations where traditional mast-based instrumentation is infeasible, will be of great value. Finally, quantifying the uncertainty of individual UAS measurements will facilitate the propagation of these uncertainties to spatial correlation and flux measurements, a capability offered by the SWUF-3D fleet.

*Data availability.* Data of the MMA and IEC mast are stored in the archive of the WiValdi wind park. Drone data will in future be stored in these archives as well, but may be made available by the authors upon request.

## Appendix A:  Flight overview

With Tab. A1, we provide a comprehensive list of which drone was hovering at which position during which flight and which sensor was fully operational during which flight. It clearly shows that some temperature sensors at specific UAS had problems for a long period of time during the campaign. Others only had occasional drop-outs. All of these failures are marked in red. Sometimes, hover periods were interrupted due to autopilot safety maneuvers (e.g. signal lost to the ground control station). Those flights are marked in light grey and excluded from the analysis as well.

## Appendix B:  Mast instrumentation

Table B1 gives an overview of the mast instrumentation that was used in this study

## Appendix C:  FWPRT calibration coefficients

The SWUF-T FWPRT sensors are calibrated individually in the EdgeTech RHCal calibration chamber. The coefficients $R_0$, $a$ and $b$ are listed in Tab. C1. The table shows all sensors that were used in the experiment. It also gives the $T_0$ bias correction that was applied from comparative measurements in the field. Sensors #9 and #25 were not calibrated in the calibration chamber individually and thus show some of the largest biases in the field. However, also sensor #1 and #4 show significant biases, which shows that the sensors are not perfectly robust and long-term stable. All sensors were calibrated within three months prior to the campaign. It can unfortunately not be traced back exactly why the two sensor show a bias. We thus certainly advice to do on-site offset calibration before a campaign.

## Appendix D:  Ventilation response test

In order to show that a small ventilation of only 1–2 m s$^{-1}$ is sufficient to prevent air from being trapped in the radiation shield housing, a small experiment was set up with convection from a teacup under the SWUF_T sensor and a small ventilator in front

**Table A1.** Table of UAS positions for each flight. The nomenclature is $zzYX$ where $zz$ is the flight height above ground in meter, $X$ the cardinal direction of the hover position relative to the MMA axis and $Y$ the cardinal direction of the hover position relative to the MMA center mast ($C$ stands for a center position). Red labels indicate temperature sensor failures.

| Flight Number ↓ / UAV number → | 32 | 12 | 13 | 14 | 15 | 21 | 22 | 23 | 24 | 25 |
|---|---|---|---|---|---|---|---|---|---|---|
| 36 | 25NW | 90NW | 90CW | 90SW | 25SW | 25NE | 90NE | 90CE | 90SE | 25SE |
| 37 | 25NW | 90NW | 90CW | 90SW | 25SW | 25NE | 90NE | 90CE | 90SE | 25SE |
| 38 | 25NW | 90NW | 90CW | 90SW | 25SW | 25NE | 90NE | 90CE | 90SE | 25SE |
| 39 | 25NW | 90NW | 90CW | 90SW | 25SW | 25NE | 90NE | 90CE | 90SE | 25SE |
| 40 | 25NW | 90NW | 90CW | 90SW | 25SW | 25NE | 90NE | 90CE | 90SE | 25SE |
| 41 | 25NW | 90NW | 90CW | 90SW | 25SW | 25NE | 90NE | 90CE | 90SE | 25SE |
| 42 | 25NW | 90NW | 90CW | 90SW | 25SW | 25NE | 90NE | 90CE | 90SE | 25SE |
| 43 | 25NW | 90NW | 90CW | 90SW | 25SW | 25NE | 90NE | 90CE | 90SE | 25SE |
| 44 | 25NW | 90NW | 90CW | 90SW | 25SW | 25NE | 90NE | 90CE | 90SE | 25SE |
| 46 | 25NW | 90NW | 90CW | 90SW | 25SW | 25NE | 90NE | 90CE | 90SE | 25SE |
| 47 | 25NW | 90NW | 90CW | 90SW | 25SW | 25NE | 90NE | 90CE | 90SE | 25SE |
| 48 | 25NW | 90NW | 90CW | 90SW | 25SW | 25NE | 90NE | 90CE | 90SE | 25SE |
| 50 | 25NW | 90NW | 90CW | 90SW | 25SW |  |  | 90CE | 90SE | 25SE |
| 51 | 25NW | 90NW | 90CW | 90SW | 25SW |  |  | 90CE | 90SE | 25SE |
| 52 | 25NW | 90NW | 90CW | 90SW | 25SW |  |  | 90CE | 90SE | 25SE |
| 74 |  |  | 25SW | 90SW | 150SW |  |  | 25SE |  | 150SE |
| 75 |  |  | 25SW | 90SW | 150SW |  |  | 25SE | 90SE | 150SE |
| 77 |  |  | 25SW | 90SW | 150SW |  |  | 25SE |  |  |
| 78 |  |  | 25SW | 90SW | 150SW |  |  | 25SE | 90SE | 150SE |

**Table B1.** Instrumentation at WiValdi that was used in this study with type and installation height.

| Mast | sensor type | installation height |
|---|---|---|
| Inflow | Thies Hygro-Thermogeber compact 1.1005.54.441 | 85 m |
| Inflow | Thies Wind Vane 4.3151.00.141 | 88 m |
| Inflow | Thies cup anemometer 4.3352.00.401 | 88 m |
| MMA | Thies Clima 3D sonic anemometer 4.3830.20.340 | 25 m, 90 m |

**Table C1.** Calibration coefficients for the sensors used in this study.

| FWPRT # | UAS # | $c_0$ | $c_1$ | $\Delta T$ |
|---|---|---|---|---|
| 1 | 23 (from 23-07-2024) | -267.66683040824887 | 5.308727476790697 | -1.8 K |
| 3 | 12 | -269.0732285831074 | 5.079828699776475 | 0.25 K |
| 4 | 25 | -267.1235152527639 | 5.677151810920186 | 1.17 K |
| 6 | 15 | -269.0721551243061 | 5.236323589105574 | 0 |
| 7 | 13 | -268.72751501503075 | 5.3329656611968534 | 0 |
| 8 | 22 | -268.3654709827349 | 5.318415986601671 | 0 |
| 9 | 24 (from 24-07-2024) | n.a. | n.a. | -1.2 K |
| 13 | 14 | -271.77055837721355 | 5.493507603639585 | 0 |
| 14 | 32 (on 22-07-2024) | -269.59953070379885 | 5.338689279389542 | 0.2 K |
| 15 | 21 | -266.6111605253554 | 5.38704463397534 | 0 |
| 17 | 23 (until 22-07-2024) | -272.1692867723404 | 5.454972064785658 | 0 |
| 18 | 24 (until 23-07-2024) | -266.8939033978739 | 5.314431828725549 | 0 |
| 25 | 32 (from 23-07-2024) | n.a. | n.a. | -3.7 K |

(see sketch in Fig. D1). At $t_0$ the ventilator is switched on and the fast response FWPRT measurements immediately respond with a quick temperature drop.

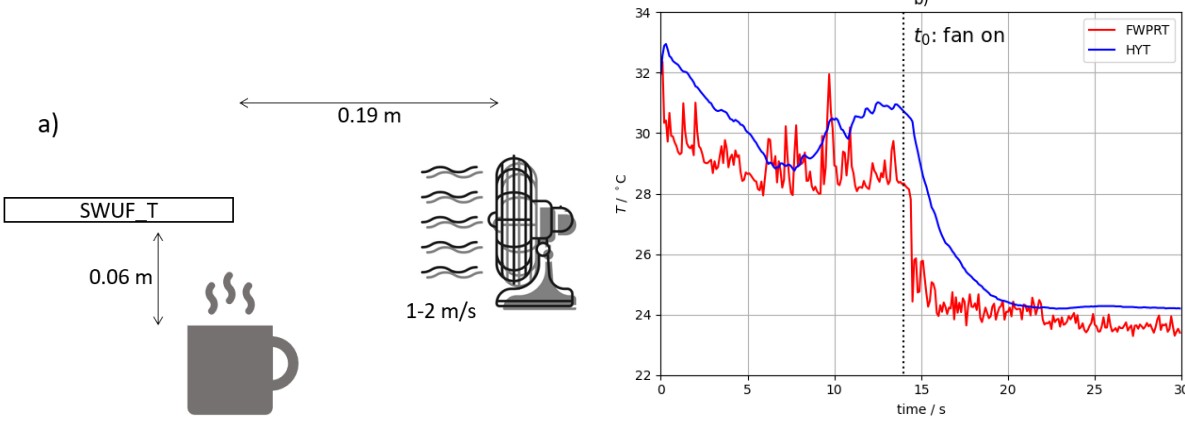

**Figure D1.** Sketch (a) and time series (b) of a lab experiment with the SWUF_T sensor. The sensor is placed over a tea cup with hot water and ventilated from a defined point in time.

 **Appendix E: Vertical velocity variance**

Vertical velocity in this study is calculated according to Wildmann and Wetz (2022). The comparison to sonic anemometers in this study gives slightly worse coefficients of determination than in the 2022 study at Falkenberg, i.e. $R^2 = 0.13\text{m}^2\text{s}^{-2}$ at WiValdi vs. $R^2 = 0.07\text{m}^2\text{s}^{-2}$ at Falkenberg. Figure E1 shows the scatter plot of two masts against each other and the UAS against the masts, showing that the variability of vertical velocity between masts is even higher than the difference between
410 UAS and mast and probably the main cause for the larger uncertainty.

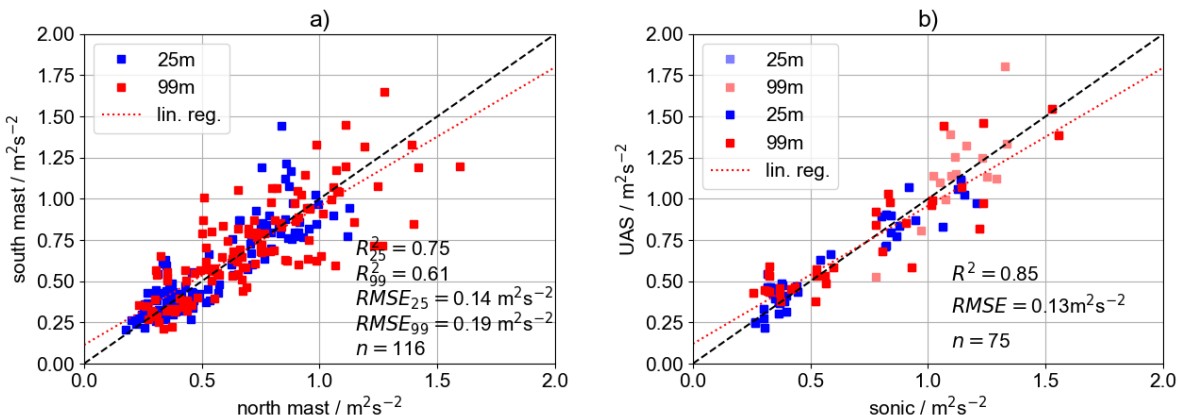

**Figure E1.** Comparison of vertical velocity variance $\sigma_w^2$ between sonic anemometers of north and south mast (a) and between UAS and the closest mast (b). Shown are the comparisons at 25 m level (blue) and 99 m level (red). The combined linear regression is shown as red dotted line.

*Author contributions.* NW designed and built the first SWUF-T sensors, did the data analyses and writing of the paper. LG contributed to the calibration of the sensors and the corresponding section in the manuscript.

*Competing interests.* We declare no competing interests.

*Acknowledgements.* This research has been supported by the HORIZON EUROPE European Research Council (grant no. 101040823,
ESTABLIS-UAS). This work was partly accomplished within the research project Deutsche Forschungsplattform für Windenergie (http://dfwind.de). We greatly acknowledge the financial support of the German Federal Ministry for Economic Affairs and Climate Action, through FKZ 0325936, that enabled this work. We acknowledge the support by the DLR-WX and DLR-UX teams in supporting the campaigns. We especially thank Johannes Kistner and Almut Alexa for their hard work in preparing and carrying out the campaigns at Krummendeich and

Cochstedt. We thank Paula Hofmann, Paul Waldmann, Michael Lichtenstern and Anke Roiger for the support with smoke plume visualization and providing the video of this test. The authors acknowledge the use of a large language model (e.g., Gemini and llama3.2) for language improvements.

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
