# Peer review of "Towards sensible heat flux measurements with fast-response fine-wire platinum resistance thermometers on small multicopter uncrewed aerial systems."

_EGUsphere, 2025_

## Author Comment (AC1)

**Towards sensible heat flux measurements with fast-response fine-wire platinum resistance thermometers on small multicopter uncrewed aerial systems.**

Norman Wildmann[1] and Laszlo Györy[1]

[1]Deutsches Zentrum für Luft- und Raumfahrt e.V., Institut für Physik der Atmosphäre, Oberpfaffenhofen, Germany

**Correspondence:** Norman Wildmann (norman.wildmann@dlr.de)

**1 Review response**

We want to thank the reviewer for this extraordinarily detailed and helpful review. It addresses many very relevant points with regards to the manuscript and directed our focus onto some very relevant points that we agreeably had not addressed well enough in the original manuscript. It also gives a lot of ideas and perspectives for further studies and analyses of the atmospheric phenomena observed in the study. It made it obvious that we also have to clearly state what is the objective of this manuscript and what is out of scope of this manuscript and needs to be addressed in future studies. Because of the extent of the review, we want to start the response with a table that summarizes the main topics from our point of view and briefly states how we address these issues in the revised manuscript. We also want to comment in this table on the relevance for the objective to qualify the suitability of fast response temperature measurements on small multicopter UAS for heat flux measurements. Detailed answers to every review comment are given below in Sections 1.1 and 1.2.

**Table 1.** Main review responses and manuscript changes

| | |
|---|---|
| Differences in analysis methods for sonic and UAS | Where it makes sense and was not the case before, the methods were unified, i.e. the same sampling rate was applied. The results showed to not be strongly dependent on the changes. |
| Time-lag correction | We applied the time-lag correction, but found it to be of minor relevance for the results because the changes in the observation periods and thus the averaging periods are comparatively small. |
| Buoyancy-flux vs. sensible heat flux | All analyses were changed to buoyancy flux, which improved the comparison. |
| Temperature sensor housing | We show that there is no significant impact by the sensor housing and ventilation is always present through a small extra experiment and some CFD simulations that had been done in a student work. |
| Temperature sensor calibration | The calibration accuracy, both in the laboratory and with in-field offset correction are well within 0.5 K in the campaign. The offset correction can cause larger errors if measurement points are more than 10 K from the temperature at which the offset calibration was done. A long-term study of calibration stability is beyond scope of this manuscript, but may be relevant in future. |
| Vertical velocity variance and its influence on heat flux | We carefully revisited the calibration coefficients of the vertical velocity calculation and could see that, as written in the manuscript, this parameter has the biggest influence on flux uncertainties. We add a short chapter in the revised manuscript that describes the method and the changes in comparison to **?**. A scatter plot of vertical velocity variance in comparison to MMA data is now also provided. |
| Differences of heat flux at the two measurement heights, footprint and atmospheric conditions | Within this manuscript for AMT we want to focus on the comparison between sonic anemometer and UAS measurements. The two heights are beneficial to get a broader range of magnitude for temperature variance and heat flux, but it is beyond the scope of this study to analyze the boundary-layer dynamics and discuss the effects of footprint and measurement height on the fluxes. With an established measurement setup and uncertainty quantification, we will tackle the ABL research questions in future publications. |

**1.1 Review General comments**

1. *Although the purpose of this manuscript is a proof-of-concept, detailed analysis is only provided for a narrow selection of data, without clearly stating why a data selection has been made. Table 1 indicates 3 calibration flights with 5 UAV each. Time series and spectra are only shown for 1 flight and 4 UAV. I acknowledge the fact that the authors are not hiding flight 69, which did not show great agreement, but at the same time, it is not even mentioned why the third calibration flight is not analyzed. All three flights should have been analyzed in the same way. On the other hand, second-order statistics, i.e., virtual temperature variance and sensible heat fluxes, are then presented for all flights, which gives a robust foundation for more solid conclusions. For completeness, I would, however, request to also include similar analyses as shown in Figure 9 and 10 for the vertical velocity variances.*

   We are sorry for not being clear and transparent on the reasons why some flights are presented and others were not. We did mention that in this early rollout of sensors to the fleet, multiple sensors had problems with missing data in part of the flights. The third calibration flight could not be used, because the sensors were not operating. It will be removed from the list of flights altogether to avoid misunderstandings. The vertical velocity variance comparison will be added to the appendix of the manuscript and is presented here in Fig. 2.

2. *The validation against the sonic anemometer data and the cross-validation of two masts, to quantify the uncertainty in the experimental setup due to heterogeneities, appears to be carried out applying different methods. These methods should be clearly documented and designed to allow for the best and most fair comparability between the different data sets. This includes cross-correlation of the different time series to eliminate the time lag resulting from spatial separation (in reasonable time windows), using the same mast data for the mast-mast validation as for the UAV-mast validation. This implies that error statistics are computed for the same set of data and not for 172 samples vs 75 samples, which are based on different methods. Don't compare the buoyancy flux from the sonic anemometer to the sensible heat flux from the UAS (this may be just an error in the terminology used). A more thorough data processing as suggested, in particular the time-lag correction, may contribute to a lower error estimate for the UAV turbulence parameters as well as for the experimental setup.*

   We should be more clear on how data is calculated and may have missed to provide the necessary information. It is not the case that different methodologies are used. We do not consider it a different methodology when different sample sizes are used. To get the background uncertainty of the experiment, we use all available ten-minute periods from the mast within the experimental periods, i.e. 22.07.2025 06:00-18:00 UTC, 23.07.2025 06:00-13:00 UTC and 25.07.2025 06:00-16:00. For all ten minute periods we calculate mean, variance and buoyancy flux. We considered this the baseline, background uncertainty of the experimental setup in time and space. We do not make a direct comparison for the single measurement points, because we believe that a broader database gives more robust results. The periods that were originally chosen are actually longer than the periods when UAS were flying, so we now reduce those times to the actual flight periods, i.e.

[Figure]

**Figure 1.** Scatter plot between temperature variance as calculated without time lag (x-axis) and with time lag (y-axis).

22.07.2025 10:00-15:30 UTC, 23.07.2025 06:30-10:00 UTC and 25.07.2025 07:30-14:00. We had not included 25.07. in the previous plots, but added them now, increasing the total number of points in the plots.

The time lag correction was mentioned by us in the conclusions as a possible additional measure and we now included it as a response to the reviewer comment. The time lag between the sensors that are 50 m separated is of the order of a few seconds. For single-point calculations of the temperature variance with ten-minute time series, the differences are very small (below 0.002 $K^2$, see Fig. 1).

**1.1.1 Sensible heat flux versus buoyancy flux**

*We understand that in the previous manuscript, it was not always very clear in terms of terminology if we calculate buoyancy flux or sensible heat flux or other flux terms. We will try to be more precise in the revision. We decided to convert all data to buoyancy flux, since the drones carry a humidity sensor that allows to approximately convert all temperature readings of the drone to sonic temperature equivalents better than vice versa. The new paragraph now reads as follows:*

Using the eddy covariance method, sensible heat flux, $H$, can be calculated from variations in temperature and vertical wind speed as

$$H = \rho C_p \overline{T'w'} \tag{1}$$

where $\rho$ is the air density, $C_p$ is the specific heat capacity of air and $\overline{T'w'}$ the covariance of temperature and vertical wind speed. Since the sonic anemometers do not directly measure $\overline{T'w'}$, but the buoyancy flux $\overline{T_s'w'}$ (Liu et al., 2001), we use this parameter throughout this study for the comparison between UAS and sonic:

$$H_b = \rho C_p \overline{T_s'w'} \tag{2}$$

Both variables $\rho$ and $C_p$ can be calculated from thermodynamic measurements made by the drone itself, namely temperature, humidity, and pressure. We apply a correction for moisture to the specific heat capacity of dry air, such that

$$C_p = 1004.67 \text{ J kg}^{-1}\text{K}^{-1} (1 + 0.84 m_h) \tag{3}$$

where $m_h$ is the water vapor mixing ratio.

**1.1.2  Sonic temperature**

The in-flight performance of the temperature sensor requires validation through field experiments. The preferred method is to fly the UAS in close proximity to sonic anemometers. A sonic anemometer provides a temperature reading derived from the speed of sound, which is often referred to the 'sonic temperature', $T_s$. This value is close to, but not equal to virtual temperature $T_v$. A common equation to calculate sonic temperature from air temperature $T$ is according to Schotanus et al. (1983):

$$T_s = (T + 273.15)(1.0 + 0.51q) \quad , \tag{4}$$

where $q$ is the specific humidity. The SWUF-T sensor also measures humidity, so that measured air temperature by the FWPRT temperature sensor can approximately be converted to an equivalent sonic temperature more easily than vice versa. Therefore, all temperature readings from the UAS that are compared with sonic anemometer measurements are converted to sonic temperature according to Eq. 4.

3. *The sonic anemometer data is sampled at a higher frequency (not clearly stated) than the UAV data and Figure 8 indicates that the smallest resolved scales are relevant. Since these scales cannot be resolved with the UAS the sonic data should be downsampled to the same frequency and use matching time windows (corrected for the time lag) as the UAS data for a direct comparison.*

   The sonic anemometer data is available at 10 Hz. UAS data was cut off at 4 Hz to remove noise in the first draft. We now changed the cut-off for the UAS to 5 Hz and downsampled the sonic anemometer measurements to the same frequency.

4. *The focus of this manuscript is on high-resolution temperature measurements and sensible heat fluxes; however, little attention is given to the measurement of the vertical velocity. A corresponding manuscript has already been published*

*demonstrating the ability of the system for the retrieval of vertical velocity fluctuations. However, it is not sufficient to only refer to it without summarizing the important details of the methods used to determine the vertical velocity, the uncertainties and limitations of this method, and potential improvements or changes to the method presented in Wildmann et al. (2022).*

We focus on the temperature measurements in this manuscript because it is an essential part and deserves the attention. We present heat flux estimates based on the eddy covariance method, which requires the vertical wind measurements as presented in Wildmann et al. (2022) with its limitations. All the details are presented and published in this reference. Further improvements are certainly possible and are ongoing, but they are out of scope of this manuscript and a separate study is in preparation for this. We did revisit the vertical velocity estimates for this study again in the meantime and adapted some parameters to the newest version of rotors and UAS weight. We include Fig. 2 in the appendix of the manuscript to show the error statistics for vertical velocity variance.

[Figure]

**Figure 2.** Comparison of vertical velocity variance $\sigma_w^2$ between sonic anemometers of north and south mast (a) and between UAS and the closest mast (b). Shown are the comparisons at 25 m level (blue) and 99 m level (red). The combined linear regression is shown as red dotted line.

These changes, along with the basic equations are included in the revised manuscript in Section 4 as follows:

**1.1.3 Vertical velocity esimate**

Attempting to calculate fluxes with the eddy covariance method requires synchronous measurements of temperature with vertical flow velocity $w$. As described in Wildmann and Wetz (2022), vertical velocity can be estimated from thrust and lift of the UAS. The force acting on the drone in $z$-direction $F_z$ in the body frame of the drone is a combination of gravitational force ($mg$ rotated into body frame with roll $\varphi$ and pitch $\theta$ angle), vertical acceleration forces $m\ddot{z}$, thrust $T$

and lift force $F_L$ which is related to the drag force $F_x$:

$$F_z = -mg\cos(\theta)\cos(\varphi) + m\ddot{z} + T + F_L(F_x) \tag{5}$$

Vertical velocity is derived based on a calibrated curve using the equations

$$w_b = \begin{cases} c_{z\uparrow}F_z^{b_{z\uparrow}} & F_z \geq 0 \\ c_{z\downarrow}F_z^{b_{z\downarrow}} & F_z < 0 \end{cases}. \tag{6}$$

Since the study in 2022, new rotors and new batteries were installed on the SWUF-3D UAS, which changes the thrust and lift behaviour. Parameters were adjusted accordingly. It also showed that it is beneficial to first rotate the forces into the geodetic coordinate system and do the vertical velocity calibration in this frame of reference, so that:

$$w = \begin{cases} c_{z\uparrow}F_z^{g_{z\uparrow}} & F_z \geq 0 \\ c_{z\downarrow}F_z^{g_{z\downarrow}} & F_z < 0 \end{cases}. \tag{7}$$

5. *The article largely relies on previous work by one of the authors, but it should refer to the peer-reviewed article Wildmann et al. (2022) and not the corresponding preprint/discussion paper.*

We apologize for this mistake, we had an old bibtex-reference and updated it now for the revised manuscript.

6. *The sensor placement in a rather narrow housing may hinder vertical motion and potentially trap air. The design of this housing needs to be clearly described and some sort of validation, e.g., CFD simulation, should be presented to verify potential flow distortion effects of the housing under different wind conditions. In particular, in free convection cases, the ambient air velocity vector can be almost vertical. It is not clear whether the housing allows for proper airflow under such conditions, so there may be a minimum wind speed limit for this method.*

Sensor placement and airflow around sensors has been studied by many researchers and we agree that it has a relevance for good measurements. We designed the housing with a minimum of flow blockage, it is open to the front and the bottom, which could not be seen in the presented pictures in the manuscript. We change it to a picture taken from a front angle (see Fig. 4). In a recent student work, CFD simulations were performed of the QAV drone at maximum thrust (maximum take-off weight), without any wind inflow (see Fig. **??**). It shows that at the location of the temperature sensor, an airflow of approximately 2 m s$^{-1}$ can still be found. This is enough to ventilate the sensor. As soon as there is a background wind, the downwash of the drone will be pushed downstream and the temperature sensor is subject to natural ventilation. A CFD-simulation of the sensor housing itself was not performed, because airflow around such a small structure requires a sophisticated meshing and simulation setup and the cost does not seem to be justified at this point.

[Figure]

[Figure]

**Figure 3.** SWUF-T board design concept (a) and pictures of the sensor without (b) and with (c) housing.

[Figure]

**Figure 4.** Front view (left) and top view (right) of a flow simulation around the QAV250 drone. The color bar shows the absolute flow speed.

We performed a small experiment to show that a minimum ventilation of 1–2 m s$^{-1}$ is sufficient to prevent air being trapped in the radiation shield. With the setup as shown in Fig. 5a, we generated convection over a teacup with hot water and placed the sensor $\approx$6 cm above the cup. A small ventilator, providing a constant airflow of $\approx 1.5$ m s$^{-1}$ was placed 0.19 m in front of the sensor and switched on at time $t_0$. Fig. 5 shows the resulting time series for both, the FWPRT and the HYT sensor. It is obvious that the FWPRT sensor immediately falls back to the background temperature as soon as the fan is switched on. The HYT shows some longer time response.

We add this experiment description to the appendix of the manuscript.

[Figure]

**Figure 5.** Sketch (a) and time series (b) of a lab experiment with the SWUF_T sensor. The sensor is placed over a tea cup with hot water and ventilated from a defined point in time.

7. *The method presented in the manuscript has great potential, but a clear statement on the motivation of measuring atmospheric turbulence with one or a fleet of small UAS is lacking.*

   The motivation to measuring atmospheric turbulence with a fleet of UAS is to be able to obtain in-situ observations of turbulence and fluxes at flexible locations where it has not been easily possible before. Such measurements allow to quantify the complexity and spatial heterogeneity of boundary-layer processes and potentially derive spatially representative fluxes. We add this motivation to the introduction of the revised manuscript.

8. *The chosen approach of using a fine wire temperature sensor and vertical velocity estimates based on the UAV state should be put into better perspective and not only compared against the fixed-wing multi-hole probe approach. Relevant publications (Fuertes et al. 2019; Greene et al. 2020; Ghirardelli, et al. 2024).*

   We agree that we should better reflect on other methods and include the suggested references into the introduction of the revised manuscript.

9. *Overall, quite a lot of data has been taken into account and the results appear robust, but the atmospheric conditions during which the experiments were carried out were rather limited. I request to a) document these conditions better, including information on atmospheric stability (no stably stratified cases or negative heat fluxes were sampled), wind speed and direction (already in Table 1), footprint and soil appearance (significant latent heat fluxes), solar radiation and cloud cover and b) take these conditions into account when interpreting the results, drawing conclusions, and potentially also for the outlook.*

   While all this information is very important to draw conclusions about ABL evolution, turbulent fluxes and energy balance, it is beyond the scope of this manuscript to analyse the reasons for different heat flux conditions. This manuscript in AMT is meant to characterize the suitability of a new temperature sensor onboard a UAS to calculate heat fluxes. It

is certainly important to understand the reasons for uncertainty in flux measurements, but we believe that it is valuable, even without knowledge of all atmospheric details, to see the comparison and evaluation of direct comparison between sonic anemometers and UAS on a small scale. It is unfortunately not possible to reconstruct soil appearance and footprint due to missing data, also solar radiation and cloud cover cannot be reproduced with good accuracy on the microscale. Stability can only be derived from a microwave radiometer on site, which is prone to errors in the very lowest layers of the atmosphere and is therefor not included. Basic information which is available (humidity, cloud cover by observation) will be added to Table 1 of the revised manuscript (see Tab. 2 here). It was premature to mention footprint as a possible reason for differences in flux measurements and this statement will be removed from the conclusions. The newest processing and calibration of 3D wind does not show these features as clearly anymore.

10. *I lack a clear statement on the limitations related to the currently limited frequency resolution. The relative contribution of small-scale turbulence to the total flux may become more important under certain conditions. What does this mean for the applicability of this method, e.g. for stable boundary layer cases?*

    In previous studies describing the capabilities of (mechanical) turbulence measurements with the SWUF-3D UAS fleet, we always mentioned that small-scale turbulence will remain difficult to measure with multirotor UAS and thus stable boundary layer research is not feasible with the current state of the art. It is true that we did not explicitly mention it this times, but it certainly should be. We add a sentence on this in the conclusions in the revised version of the manuscript.

11. *Some results are not presented consistently. Fig 6, 9, and 10 use different marker colours to distinguish different height levels, but the error statistics are presented as a bulk. The discussion of these figures sometimes lacks any discussion of differences between the measurement heights.*

    We work under the assumption that we can determine an overall uncertainty of the sensor, independent of measurement height. It helped to have the different heights in this study, since wind speed, temperature variance and turbulent fluxes differ with height and thus a broader range of conditions are observed. From our point of view it helps the understanding to see the different heights marked in different colors, but there is no benefit in distinguishing the two heights in the error statistics. It may even be misleading, because the differences that can be observed is not caused by the different height per se, but by the different intensity of turbulence that happen to occur at those heights.

12. *Some Figures are hard to read and need to be improved, including a more detailed description of what is shown.*

    Every Figure is revisited in the revised manuscript. The specifically addressed Figures are shown in this response in their new version. All captions are expended in the revised manuscript to describe the figures more completely.

**1.2 Review specific comments**

1. *L8-9: I slightly disagree with the statement that this is the first time sensible heat fluxes have been measured using multicopter UAS (e.g., Fuertes et al. 2019; Greene et al. 2020; Ghirardelli, et al. 2024). I agree that this is the first time it has been done with the method presented in this manuscript or more generally, applying the EC method from small UAS.*

These are great references that we happily include in the introduction. We had already cited Ghirardelli for the study of flow influence, and will now also mention it as an alternative to put an external sensor on a UAS. We were not fully aware of the other studies and thank the reviewer for pointing us into this direction. **?** did show a system that is capable of measuring sensible heat flux, but did not actually make a validation of it, at least in this paper. The method by Greene et al. (2022) is using vertical profiles and a gradient approach which is quite different from the approach we present, but should be mentioned as an alternative. We add corresponding statements in the introduction of the revised manuscript.

2. *L11: Since you show that the method works for wind speeds up to 8 m/s, I recommend stating this threshold specifically instead of using the term "low wind speed conditions".*

We add the number in parentheses behind the statement.

3. *Although it is stated that this study focuses on small multicopter UAS, it still deserves some attention that EC measurements can be performed from large UAS using standard sensors.*

We add the references Ghirardelli et al. (2023) and Thielicke et al. (2021) as examples for such concepts to the revised manuscript.

4. *I suggest first providing some background on the EC method and other relevant methods for estimating turbulence from UAS. The parameters (and their resolution) needed to measure turbulence using the EC method should be clearly stated, as well as the different sensors and methods for retrieving these parameters, including a critical evaluation of the different approaches for estimating turbulence, particularly the sensible heat flux and the difference between the buoyancy and sensible heat flux. It should also be made very clear that the described method does not rely on a dedicated wind sensor.*

A comprehensive description of the state of the art of the eddy covariance method and alternatives is out of scope of this manuscript and we refer to Baldocchi et al. (2001) and Mauder and Foken (2011) for details on the requirements and the necessary processing steps for accurate flux measurements with the eddy covariance method. Billesbach et al. (2024) gives a nice comparison of the EC method to other methods and quantifies the related uncertainties. In our study, it is not the primary goal to estimate accurate fluxes, but to get a good comparison between UAS and sonic anemometers for the raw measurements of covariances and the directly derived flux values. We stress this in the revised manuscript.

5. *L29-30: It is not clear that this refers to resolving mechanical turbulence by making use of the UAV's INS data.*

The statements we make are very general. Turbulence measurements by any means depend on size, weight and flow disturbance of the aircraft system. We give sensor noise as an additional factor in our system, but it actually will refer to most other methods we can think of.

6. *Section 2: This section should also include important details on the UAV and the different versions used.*

We agree that we did not sufficiently describe the UAS itself and add a section at the beginning of Section 2 about the UAS. We only use one version of UAS in this study. This was probably a misunderstanding, because we mentioned the other UAS that was operated at Cochstedt. We do not show any data from that other UAS on purpose, because it would

220        not be directly comparable.

7. *L57: Maybe a typo in "MAX38165" - could be "MAX31865".*

   Yes, that is correct, we apologize for the mistake.

8. *L59: Does the conversion to temperature also depend on the length and diameter of the wire? I assume this is one of the*
225   *main reasons why this setup still requires calibration.*

   The resistance depends on length and diameter of the wire as described in Sect. 4.4.

9. *L61: Also indicate the Arduino board in Figure 1. The purpose of this microcontroller board is not clear to me. The I2C*
   *and SPI could be directly wired to the flight controller and logged with some custom script.*

   This is a very technical and interesting question and the reasons are rather practical. There were versions when we
230   connected I2C and SPI to the autopilot (interfaces on the board still exist), but this required programming sensor drivers
   into the autopilot code. Having a MAVLink message (or an ASCII output through the UART interface) created in the
   Arduino microprocessor makes the system more easily portable to other versions of autopilot software (even Ardupilot)
   and potentially compatible with other systems that use the MAVLink protocol.

10. *L65-67: A more detailed sketch of the housing would be relevant. It looks very narrow, and it is not clear where the air*
235   *inlets and outlets are. Since the purpose of the presented setup is to measure sensible heat fluxes, and you restrict yourself*
   *to rather low wind speeds, it would be very important to obstruct the vertical flow as little as possible. The housing looks*
   *like it is only allowing for horizontal flow. As shown in Wildmann et al. (2013), in most cases, radiation shields are not*
   *necessary for wires with very low thermal inertia, so why do you still use a radiation shield? In case the housing does*
   *not allow for vertical flow passing through it should be mentioned that strong convection may not be measured correctly*
240   *and you should provide a minimum wind speed or angle of attack.*

   We added the Arduino to the sketch. As written in the answer to the general comment, airflow through the sensing
   element is given in any flight condition. Within the scales that can realistically be measured with the multirotor UAS, i.e.
   1-2 Hz, corresponding to eddy sizes of few meters, we do not see the problem of air being blocked and trapped. Even at
   "low" wind speeds that can occur, the airflow is hardly only vertical.
245   The radiation shield is primarily for the HYT sensor, which is more easily affected by radiation. The tests for radiation
   errors on fine-wires in Wildmann et al. (2013) were done at an airspeed of 20 m s−1 for fixed-wing UAS. At lower
   wind speeds forced convection on the wires is smaller and radiation errors and self-heating could become slightly more
   relevant.
   It could be possible to change the sensor design in future to have the fine wires outside the radiation shield and only the
250   HYT inside, but we decided to have both sensors as close as possible to each other. Also, even better ventilation could
   be allowed with some trenches in the housing, but this would make it less protected to light rain.
   We want to emphasize that for a scalable solution for a fleet of many (dozens) UAS, it is important to keep things simple,

lightweight and effective. There is a risk of over-engineering a solution to requirements that are beyond what can be achieved within the overall setup of the multirotor UAS.

11. *L70: What is the minimum wind speed required for the wind vane mode to work properly?*

The weather vane mode is working if a roll angle larger than $1°$ can be inflicted, which is the case for wind speeds larger than $\approx 2\ \mathrm{m\ s^{-1}}$ in spanwise direction. The wind algorithm for the SWUF-3D takes spanwise wind into consideration if the UAS is not perfectly aligned. In Kistner et al. (2024) it is described how much the uncertainties increase for sideslip angles up to $30°$.

12. *L71: Replace "sensitive elements" with "sensing elements".*

Thanks, we correct the expression in the revised manuscript.

13. *L80: "The analyses in this study are based..."*

Thanks, we correct the expression in the revised manuscript.

14. *L84-85: Provide a more detailed description of the MMA pattern. From Figure 2 and Table 1, I can only guess that you aimed to sample at each of the 4 positions marked on the map at 25 m and 90 m. There are probably practical reasons why you could not always operate 8 UAV at the same time, but in flight 41, you indicate 9 UAVs. Where did the last one fly?*

We apologize for being imprecise on the pattern and the reasons why not always all UAS are available. We mention throughout the manuscript that temperature sensors sometimes had dropouts and errors. This was mainly due to a bad I2C-implementation in the Arduino-library and has been fixed in the meantime, but during the campaign this caused a lot of flights to be not usable for the temperature measurements, while wind measurements are still good for those flights. We will make this clear in the revised manuscript. We change the tables completely. Table 2 will give the meteorological conditions of the flights and the number of used UAS during the flights. Table 3 gives a detailed overview of which UAS was hovering at which position during which flight and its availability for temperature measurements. It clearly shows that some temperature sensors at specific UAS had problems for a long period of time during the campaign. Others only had occasional drop-outs. All of these failures are marked in red. Sometimes, hover periods were interrupted due to autopilot safety maneuvers (e.g. signal lost to the ground control station). Those flights are marked in light grey and excluded from the analysis. We hope that this table helps to make the availability of sensors and the size of the dataset more transparent.

We also provide an additional 3D plot to show the hover positions (Fig. 6). The 9th drone is next to the center mast at 90 m. Since the center mast and UAS#23 did not provide reliable data for most of the campaign, we removed these flights from the analyses. We will also add a 3D-plot of drone positions as shown in Fig. 6.

15. *L92-94: I am confused by this statement as it is not clear whether you used the old or new version of the UAV in this study. Can you also list the flight up to 300m in Table 1? Section 2 does not include detailed information on the UAV, neither on the new nor old version.*

[Figure]

**Figure 6.** Three-dimensional visualization of MMA flight pattern with corresponding UAS numbers.

**Table 2.** Measurement flights from 22-25 July 2024

| #  | t UTC            | pattern | z m    | Ψ deg | U m s$^{-1}$ | TI % | T °C | φ % | clouds 1/8 |
|----|------------------|---------|--------|-------|--------------|------|------|-----|------------|
| 36 | 22.07.2024 10:10 | MMA     | 25, 90 | 303   | 8.4          | 19   | 16.7 | 67  | 8/8        |
| 37 | 22.07.2024 10:50 | MMA     | 25, 90 | 300   | 8.2          | 18.4 | 16.8 | 66  | 7/8        |
| 38 | 22.07.2024 11:30 | MMA     | 25, 90 | 303   | 8.0          | 18.7 | 16.9 | 71  | 6/8        |
| 39 | 22.07.2024 12:30 | MMA     | 25, 90 | 303   | 7.9          | 20.4 | 17.1 | 74  | 4/8        |
| 40 | 22.07.2024 12:55 | MMA     | 25, 90 | 300   | 8.1          | 20.6 | 17.2 | 73  | 4/8        |
| 41 | 22.07.2024 13:30 | MMA     | 25, 90 | 292   | 8.5          | 18.8 | 17.3 | 69  | 4/8        |
| 42 | 22.07.2024 14:15 | MMA     | 25, 90 | 309   | 7.8          | 17.4 | 17.3 | 70  | 4/8        |
| 43 | 22.07.2024 14:50 | MMA     | 25, 90 | 312   | 7.5          | 18.6 | 17.4 | 68  | 2/8        |
| 44 | 22.07.2024 15:15 | MMA     | 25, 90 | 305   | 6.6          | 19.8 | 17.4 | 68  | 1/8        |
| 46 | 23.07.2024 06:40 | MMA     | 25, 90 | 228   | 6.6          | 15.7 | 19.0 | 82  | 7/8        |
| 47 | 23.07.2024 07:20 | MMA     | 25, 90 | 246   | 5.8          | 14.5 | 19.2 | 81  | 7/8        |
| 48 | 23.07.2024 07:55 | MMA     | 25, 90 | 256   | 5.1          | 21.4 | 19.4 | 80  | 7/8        |
| 50 | 23.07.2024 08:40 | MMA     | 25, 90 | 247   | 6.2          | 16.5 | 19.7 | 78  | 7/8        |
| 51 | 23.07.2024 09:20 | MMA     | 25, 90 | 243   | 5.4          | 18.4 | 19.7 | 77  | 8/8        |
| 52 | 23.07.2024 09:45 | MMA     | 25, 90 | 238   | 7.1          | 14.0 | 19.9 | 75  | 8/8        |
| 69 | 25.07.2024 07:45 | calib   | 99     | 224   | 3.9          | 19.8 | 15.3 | 75  | 7/8        |
| 70 | 25.07.2024 08:15 | calib   | 99     | 222   | 4.0          | 22.2 | 15.8 | 72  | 7/8        |
| 74 | 25.07.2024 11:55 | MMA     | 25, 90 | 212   | 5.0          | 15.1 | 19.8 | 48  | 7/8        |
| 77 | 25.07.2024 13:00 | MMA     | 25, 90 | 204   | 2.9          | 27.9 | 20.5 | 47  | 7/8        |
| 78 | 25.07.2024 13:30 | MMA     | 25, 90 | 214.5 | 3.6          | 21.8 | 21.2 | 41  | 7/8        |

To be consistent within this study, we only used the old version that was also used at Krummendeich. We maybe should not even have mentioned the new one to avoid the confusion, but we think it is relevant to explain why flights to higher altitudes which are possible and were conducted at Cochstedt are not used in the study. We prefer to not put the flight in Table 1, because that table specifically described the Krummendeich flights. The single flight at Cochstedt is self-explained by the vertical profile and does not go into the statistical analysis and validation.

16. *Table 1: I think the table should be cross-referenced somewhere in section 3. Provide a description of the different columns in the caption. Consider using "MMA" instead of "mast array". I assume the meteorological conditions are from one of the masts. Which one and which observation level? How are they averaged? Consider sorting the UAS by their IDs.*

The table shall provide a rough information of background conditions during the experiment. Data from the inflow mast is used at 89 m height above ground. The closest 10-minute average is used for wind speed, wind direction, temperature, TI and relative humidity. Cloud cover will be added based on field observations.

**Table 3.** Table of UAS positions for each flight. The nomenclature is $zzYX$ where $zz$ is the flight height above ground in meter, $X$ the cardinal direction of the hover position relative to the MMA axis and $Y$ the cardinal direction of the hover position relative to the MMA center mast ($C$ stands for a center position).

| Flight Number ↓ / UAV number → | 32 | 12 | 13 | 14 | 15 | 21 | 22 | 23 | 24 | 25 |
|---|---|---|---|---|---|---|---|---|---|---|
| 36 | 25NW | 90NW | 90CW | 90SW | 25SW | 25NE | 90NE | 90CE | 90SE | 25SE |
| 37 | 25NW | 90NW | 90CW | 90SW | 25SW | 25NE | 90NE | 90CE | 90SE | 25SE |
| 38 | 25NW | 90NW | 90CW | 90SW | 25SW | 25NE | 90NE | 90CE | 90SE | 25SE |
| 39 | 25NW | 90NW | 90CW | 90SW | 25SW | 25NE | 90NE | 90CE | 90SE | 25SE |
| 40 | 25NW | 90NW | 90CW | 90SW | 25SW | 25NE | 90NE | 90CE | 90SE | 25SE |
| 41 | 25NW | 90NW | 90CW | 90SW | 25SW | 25NE | 90NE | 90CE | 90SE | 25SE |
| 42 | 25NW | 90NW | 90CW | 90SW | 25SW | 25NE | 90NE | 90CE | 90SE | 25SE |
| 43 | 25NW | 90NW | 90CW | 90SW | 25SW | 25NE | 90NE | 90CE | 90SE | 25SE |
| 44 | 25NW | 90NW | 90CW | 90SW | 25SW | 25NE | 90NE | 90CE | 90SE | 25SE |
| 46 | 25NW | 90NW | 90CW | 90SW | 25SW | 25NE | 90NE | 90CE | 90SE | 25SE |
| 47 | 25NW | 90NW | 90CW | 90SW | 25SW | 25NE | 90NE | 90CE | 90SE | 25SE |
| 48 | 25NW | 90NW | 90CW | 90SW | 25SW | 25NE | 90NE | 90CE | 90SE | 25SE |
| 50 | 25NW | 90NW | 90CW | 90SW | 25SW |  |  | 90CE | 90SE | 25SE |
| 51 | 25NW | 90NW | 90CW | 90SW | 25SW |  |  | 90CE | 90SE | 25SE |
| 52 | 25NW | 90NW | 90CW | 90SW | 25SW |  |  | 90CE | 90SE | 25SE |
| 74 |  |  | 25SW | 90SW | 150SW |  |  | 25SE |  | 150SE |
| 75 |  |  | 25SW | 90SW | 150SW |  |  | 25SE | 90SE | 150SE |
| 77 |  |  | 25SW | 90SW | 150SW |  |  | 25SE |  |  |
| 78 |  |  | 25SW | 90SW | 150SW |  |  | 25SE | 90SE | 150SE |

17. *Table 2: Are the names related to a specific model version? If yes, the differences should be described; otherwise, this column is rather irrelevant and could be dropped. How is delta Ts computed? Please provide information on the sampling rate/frequency of the sonic anemometers.*

The names identify the location and type of sensor in the WiValdi data management system and could thus be relevant for future reference, however, the ID also is a unique identifier so that the name can indeed be dropped. The sonic anemometers are set to a sampling frequency of 10 Hz.

18. *L105: Sonic temperature is very close to but not equivalent to virtual temperature (also in L121).*

We revise the whole section to give a more precise definition of temperatures and their conversion. See response to general comments.

19. *L105: According to the manual (https://www.thiesclima.com/db/dnl/4.383x.xx.xxx_US-Anemometer-3D_e.pdf), the cross-wind correction can be enabled in the settings (TC = 1). Are you sure that the data is not already cross-wind corrected?*

Thank you for this finding. We double-checked and indeed, the cross-wind correction is enabled in the sonic anemometers. We revise our statement in the manuscript accordingly.

20. *L109-117: Make a clear difference between sonic temperature and temperature.*

We revise the whole section to give a more precise definition of temperatures and their conversion. See response to general comments.

21. *L106-107: Do the Thies 3D sonic anemometers provide temperature output along orthogonal axes u, v, w or along the non-orthogonal axes given by the transducer paths? I found a corresponding statement in the manual, but it doesn't make any sense to me to also rotate the temperature data into an orthogonal coordinate system.*

Sections 2.1 and 2.2 of the manual state that the $u$, $v$, $w$ directions are along the transducer paths. There is no rotation performed on the temperature outputs.

22. *L111: Also provide the sensor details on the inflow mast to make it comparable to the instrumentation on the other masts.*

Inflow (IEC) mast instrumentation that is used in this study:

– Temperature and humidity at 85 m: Thies Hygro-Thermogeber compact 1.1005.54.441

– Wind vane at 88 m: Thies Wind Vane 4.3151.00.141

– Cup anemometer at 88 m: Thies cup anemometer 4.3352.00.401

We add a table in the Appendix of the manuscript that gives this information.

23. *Figure 3: Add a list explaining the error statistics provided in the figure. Indicate temperature in the axes labels, e.g., Ts_MMA and Ts_IEC. What does IEC stand for? To which mast do the parameters indicated as text refer? It would be better to provide these error statistics for each mast. Why is the data from the central mast and the other heights not shown here?*

After reconsideration, we believe that the figure does not add any relevant information to the manuscript. It is well known that absolute temperature readings from sonic anemometers are not trustworthy. Absolute temperatures are also not used and needed for the comparison of variances and heat fluxes. We therefor remove the figure and rephrase the text. "IEC" is an internal name for the mast, because it is meant to be the "International Electrotechnical Commission" standard reference mast that can be used for certification measurements. We should have explained this and will not use the term any more in the revised manuscript.

24. *T_s and T_v are not equal, but a more direct comparison to T_s could be achieved by simply modifying Eq5, by changing 0.61 to 0.51 you compute T_s instead of T_v.*

We revise the whole section to give a more precise definition of temperatures and their conversion. See response to general comments.

25. *E4: What is r_v? Also, mixing ratio (m_h)?*

340     We revise the whole section to give a more precise definition of temperatures and their conversion. See response to general comments.

26. *The difference between the sensible heat flux and the buoyancy flux ( ) cannot be ignored and requires some discussion, e.g., estimating the difference for relevant conditions. It should also be made clear that the covariance of w and T_s is close to proportional to the buoyancy flux.*

345     We revise the whole section to give a more precise definition of temperatures, heat fluxes and their conversion. See response to general comments.

27. *L137: m_h is already defined.*

        We revise the whole section to give a more precise definition of temperatures, heat fluxes and their conversion. See response to general comments.

350 28. *L143: This is rather an input for the outlook: Could longer measurement periods be achieved by "stitching" time series from different UAVs together, i.e., one drone replacing another after 10 minutes or so?*

        This could be done and we had done it before as shown in **?**. The operational resources however significantly increase and could not be made available in that campaign. At the moment, we put more effort into increasing flight times with the individual drones. Newer batteries already allow flight times up to 25 minutes with the QAV drones, a new airframe

355     allows flight times over 60 minutes at the cost of stability in high wind speed and turbulence. Stitching time series from multiple sets of drones remains a possibility if longer, seamless time series are necessary.

29. *Eq8: It should be mentioned that R0, A and B are the coefficients you determine in the calibration experiment. For clarity, it may also be beneficial to use lowercase letters for these coefficients.*

        We change the variable names to lower case $a$ and $b$. However, these are not directly the coefficients that are determined

360     in calibration. As described, we apply an even simpler first-order polynomial for the temperature range we operate in, i.e. $10 \ldots 50°$C. The section is revised accordingly.

30. *Fig4: It would be great to also show the parameters (R0, A, B) determined in this calibration experiment, e.g., in the form of a table or directly in the legend (move the legend to the side of the figure). It would also be very interesting to see whether there is any drift. Have these calibration experiments been repeated at some point for some of the sensors, or*

365     *is it maybe possible to do this now? Alternatively, the stability could be determined based on a comparison to the HYT sensors, preferably over a longer time period than the 4 days covered in this manuscript. This would, however, require that the HYT sensors have acceptable long-term stability.*

        This is certainly an interesting topic that could be investigated in a future study, but is out of scope of this study. During the experiment, we did not see any drift or instability for the temperature sensors that are used within their given

370     uncertainties. The HYT temperature sensor is also very stable, it is mostly the humidity sensor which is very susceptible to contamination, dirt and damage. We give Tab. 4 in the revised manuscript that shows all calibration coefficients of

**Table 4.** Calibration coefficients for the sensors used in this study.

| FWPRT # | UAS # | $c_0$ | $c_1$ | $\Delta T$ |
|---------|-------|-------|-------|------------|
| 1 | 23 (from 23-07-2024) | -267.66683040824887 | 5.308727476790697 | -1.8 K |
| 3 | 12 | -269.0732285831074 | 5.079828699776475 | 0.25 K |
| 4 | 25 | -267.1235152527639 | 5.677151810920186 | 1.17 K |
| 6 | 15 | -269.0721551243061 | 5.236323589105574 | 0 |
| 7 | 13 | -268.72751501503075 | 5.3329656611968534 | 0 |
| 8 | 22 | -268.3654709827349 | 5.318415986601671 | 0 |
| 9 | 24 (from 24-07-2024) | n.a. | n.a. | -1.2 K |
| 13 | 14 | -271.77055837721355 | 5.493507603639585 | 0 |
| 14 | 32 (on 22-07-2024) | -269.59953070379885 | 5.338689279389542 | 0.2 K |
| 15 | 21 | -266.6111605253554 | 5.38704463397534 | 0 |
| 17 | 23 (until 22-07-2024) | -272.1692867723404 | 5.454972064785658 | 0 |
| 18 | 24 (until 23-07-2024) | -266.8939033978739 | 5.314431828725549 | 0 |
| 25 | 32 (from 23-07-2024) | n.a. | n.a. | -3.7 K |

the sensors that were used in the experiment. It also gives the $T_0$ bias correction that was applied from comparative measurements in the field. Sensors #9 and #25 were not calibrated in the calibration chamber individually and thus show some of the largest biases in the field. However, also sensor #1 and #4 show significant biases, which shows that the sensors are not perfectly robust and long-term stable. All sensors were calibrated within 3 months prior to the campaign. It can unfortunately not be traced back exactly why the two sensor show a bias. We thus certainly advice to do on-site offset calibration before a campaign.

31. *L155-165: This section should be linked better to Eq 8. You start with a second-order polynomial fit and determine that B tends to 0, so you can use a simpler linear model. A bias is accounted for by adjusting R0, and the slope coefficient is A.*

The whole paragraph will be rephrased to better explain the calibration effort:

Due to the temperature limitations of the EdgeTech RHCal calibration chamber, we cannot measure $R_0$ directly. Instead, we fit the measured resistance values to the known calibration temperatures with a first-order polynomial:

$$T_{\mathrm{pt}} = c_1 \cdot R_{\mathrm{pt}} + c_0 \ . \tag{8}$$

We obtain two coefficients $c_0$ and $c_1$ for each FWPRT sensor (see Tab. 4). These coefficients are then used to calculate the temperature $T_{\mathrm{pt}}$ based on the measured resistance $R_{\mathrm{pt}}$ of the platinum fine wire.

32. *L165: Instead of providing an average abs RMSE, it would be beneficial to provide uncertainty estimates for both the 0D and 1D calibrated sensors. Following the purpose of this manuscript, it would be of high value to demonstrate the value*

*of a higher calibration effort. This, in combination with tracing the individual sensors, may also be valuable information*

390 *for the interpretation of the results.*

We agree that we can make this difference more transparent. When calibration with the first-order polynomial is done and we assume no degradation or human errors, the RMSE will be very small. A determination of the RMSE against the calibration chamber temperatures after the polynomial fit yields 0.033 K. If we use the same calibration coefficients for all sensors that were shown in Fig. 4 of the manuscript and only do a bias correction at a single point (e.g. 25°C), the

395 RMSE calculated over the whole calibration range goes down to 0.46 K. Fig. 7 illustrates the error over the calibration range. It shows that within ±10 K of the offset-calibration point, the maximum errors are below 0.5 K, but it also shows that they can be larger outside of that range.

[Figure]

**Figure 7.** Calibration error of 5 FWPRT sensors with respect to the reference sensor in the calibration chamber after 0 D calibration (bias correction at 25°C, red curves) and 1 D calibration (grey curves).

33. *L168: For clarity, you should use the term mechanical turbulence.*

We agree that we should be more specific which turbulence variables were studied in the references. Introducing the term

400 'mechanical turbulence' could be misleading, if it is interpreted as the production mechanism of turbulence (NOAA for example defines mechanical turbulence as "Created by topographic obstacles in flow"). The turbulence analyzed from 3D wind components in previous studies could also be thermally produced. We thus rephrase to "the resolution of turbulence by measuring the three-dimensional wind vector ...".

34. *The presentation of the results should include more background information on the atmospheric conditions during these*

405 *3 calibration flights, the setup of the experiment, and what is shown in the figures. A few suggestions and questions that should be answered here: Repeat the date and time of the flights in the text, refer to Table 1. From Table 1, I get that all UAVs were flown at 99m, so why are there data points for 25m in Fig6b? The three flights were completed within a period of less than 1.5h. How did the atmospheric conditions change during this period? How about solar radiation*

*and atmospheric stability? How do you get to around 30 data points when using 3 flights with 4 UAVs? Can you add the information on which sensor was used and how they were calibrated? It would be interesting to see whether it is possible to detect the effect of the higher calibration effort for some of them (the temperature range covered here may be too narrow).*

There are some misunderstandings for this analysis and we realize that we did not explain it sufficiently. The direct answers to the questions are:

- Fig.6b does not show the calibration flight, but all flights in proximity to the MMA. They should be separated and Fig.6b should be explained separately, in comparison to Fig.3. However, since we think it is best to shift the focus away from absolute temperature readings, we will remove this plot from the revised manuscript.

- We only use two of the calibration flights (flight #60 and #70). In flight #71, temperature sensors were not operating. These calibration flights were actually primarily performed for wind calibration and only happen to be useful for temperature sensor inter-comparison. We will remove flight #71 from the list of flights. The flights were done in the morning when an increase of temperature during the time period could be observed according to the values listed in Tab. 1. A slight increase of turbulence could also be observed.

- There are no 30 data points for the calibration flight. Fig.6b shows all flights, including MMA pattern and from the 4 UAVs we only get 4 measurements as presented in Sect. 5.4.1.

- The experimental uncertainties are much larger than the calibration uncertainties. Systematic differences between calibration methods can not be detected during these flights.

In the revised manuscript, these changes are incorporated.

35. *Fig 6a: The figure is not clear enough, since it is not possible to distinguish individual lines. It is not even possible to see whether it is 2 or 4 UAV temperature curves or whether one of them has a larger variability than the other. Consider using a stacked figure layout or at least apply offsets of e.g. 0.25K increments to the different lines and use different colours.*

The purpose of this plot is to show that the lines are on top of each other. A separation with offsets makes it harder to see this in our opinion. We do give the curves different shades of color to give a better idea of the spread as shown in Fig. 8. We also include the questionable readings by QAV21 for reference.

36. *Fig 6b: Label the axis correctly (T_sonic, T_FWPRT), correct the units (K), use different colours/symbols for the different UAS/FWPRT. The caption should indicate the label (b) and state the closest mast is south.*

We remove this figure from the revised manuscript. It was somewhat misplaced and does not add much information. The revised manuscript's focus will be more directed to the questions that were raised in this review.

37. *L185: Do you mean below 0.1 K?*

Yes, of course, sorry for the mistake that will be corrected in the revised manuscript.

[Figure]

**Figure 8.** Time series of virtual temperature for five UAS (red) in close proximity to the met mast array and corresponding sonic anemometer readings (blue) for flight #69 (a) and #70 (b).

38. *L193: I agree that fast sensors are preferable for soundings, but they also have to be accurate over a wide temperature range, have good stability and robustness.*

    This is certainly the case, especially if they are deployed in an operational framework. For a research setting, we can accept individual calibration efforts and a replacement if a sensor breaks. We think it is important to report on the benefits from a research perspective, even though more work would be necessary to make a robust product for operational use.

39. *It would be relevant to provide more details on the response time correction applied, e.g., the time constant used, the function applied, and filtering to avoid the amplification of noise. How does the applied time constant compare to the one stated by the manufacturer? The large relative difference between ascent and descent at the top of the profile for the FWPRT in Fig7b deserves a more detailed interpretation. How long was the UAV hovering at this altitude before starting its descent? Could this be an artefact resulting from a wrong altitude measured by the barometer due to reduced thrust when transitioning from ascent to descent?*

    The time constant for the HYT sensor showed to be very large in our case. We had to set $\tau=20$ s to get completely remove the hysteresis between ascent and descent. This is much longer than the specified time constant ($< 5$ s) We filtered with a moving average of 5 s.

    The difference between ascent and descent at the top of the profile at WiValdi cannot be explained with a wrong altitude. Figure 9 shows the GPS and barometric altitude in comparison, which match very well. The used altitude is a filtered solution that uses both pieces of information. The time between ascent and descent in this case is ten minutes which explains the different temperature that has evolved during the morning transition of the ABL with an eroding temperature inversion exactly during that period. We will explain this in the revised manuscript.

[Figure]

**Figure 9.** GPS (blue) and barometric (orange) height for the vertical profile during flight # 20.

460    40. *Section 5.4.1: The section header is a bit misleading since the section also covers the comparison against the sonic anemometers. I recommend focusing on the validation of the calibration flights and including all three of them.*

We rename the section title to: "Calibration and inter-drone comparison". We can unfortunately not include all three calibration flights, because only two had a sufficient number of working temperature sensors. Both of these flights will be shown as in Fig. 8 here.

465    41. *L222-223: The interpretation of flight #69 is unfortunately a bit shallow. From Table 1, I would expect similar conditions between all three calibration flights. From TI for #69, I would expect the vertical velocity variance measured by the sonic to be in the order of 0.77. A value of 0.3 indicates highly non-ergodic turbulence or some other problem, potentially flow distortion by the mast. Why is flight 71 not shown?*

There is certainly a difference in ABL conditions between flight #69 and flight #70. This can also be seen in Fig. 8.

470    The value for TI in Table 1 only gives a rough background value as measured by the inflow mast approximately 600 m upstream. The calibration flights were done in a transitional period. The turbulence intensity on the inflow mast increased slightly prior to the MMA location.

42. *Figure 8: I have problems seeing whether it is three or four lines for the SWUF-T. Can you use different colours and labels also indicating the date and sensor ID for the same type of sensor?*

475    It is not the purpose of this plot that every line can be followed individually and in detail. We do not think that this could be done appropriately even with different colors and labels. This would overwhelm the plot. The message is that all "five" lines are close to on top of each other. Please be aware that due to the change of calculation of virtual temperature and sonic temperature according to the response to the general comment, the spectra have changed somewhat and actually better align now between sonic and UAS as can be seen in Fig. 10. The date and time is of flight #70.

[Figure]

**Figure 10.** Spectra of sonic temperature for UAS flight #70 next to a sonic anemometer showing measurements of the sonic (blue), the slow HYT271 (green) sensor and the FWPRT (red).

480    43. *L232: For clarity, you should add "at the corresponding height level" or similar.*

We add this in the revised manuscript.

44. *L234: R2 is not the correlation coefficient (correct further up but also wrong further down).*

We apologize for this wrong statement, of course $R^2$ is the coefficient of determination, which is the square of the correlation coefficient $R$ and we will correct this throughout the manuscript.

485    45. *L236-237: In Figure 6a, it looks like there is a clear time lag between the UAV-based and sonic-based temperature signals, even at a TI value of 22.2. Even if it does not eliminate the error induced by this spatiotemporal separation, there is a good chance to improve your agreement by determining the time lag from cross-correlation analyses. This could also have the nice side effect that you could compare time series directly and compute correlation coefficients for the instantaneous time series. The sonic data would, of course, have to be downsampled to the same frequency as the*
490    *UAS.*

As described in response to the general comments, we did implement a correction for advection, but it did not change the results significantly.

46. *Although the error statistics in Fig 9 don't distinguish between the two heights shown (I think they should), Figure 9b indicates that there is a larger scatter at 99m compared to 25m, which is somewhat surprising. This should also be*
495    *discussed.*

We cannot confirm this statement. There is no significant difference in scatter between 99m and 25m. There is just more

points in a certain range of variances for 99m. For a sensor validation, we do not think that it is necessary and helpful to distinguish the coefficient of determination and RMSE between the heights. We want to show the overall values.

47. *L245: Use a different notation to express numerical ranges, e.g., "2 m to 5 m".*

The AMT submission guideline explicitly states that "A range of numbers should be specified as "a to b" or "a...b". The expression "a–b" is only acceptable in cases where no confusion with "a minus b" is possible." We consider the dots a good notation for numerical ranges.

48. *L249: This statement should include a reference.*

We include the reference Wildmann and Wetz (2022) in the revised manuscript at this point.

49. *L250-251: As mentioned, the 10min sampling period is rather short and may thus be subject to large uncertainties due to poor statistical representation of larger eddies. Non-stationary is often, but not necessarily, related to too long averaging periods.*

Yes that is true, both can be the case.

50. *L255-260: Both correlations, sonic-sonic and UAV-sonic, could be improved when correcting for the time lag. Given the fact that the masts are separated by 100m and the UAVs distance to the reference sonic is around 50m, the comparison would also become fairer. The number of samples in Fig. 10a and b is very different. Why is this the case and how does this affect the error statistics? For a fair comparison, you should re-compute the error statistics for the same periods as shown in Figure 10b. You should also downsample the sonic anemometer data, since the scales >4 Hz still carry some energy, but they cannot be resolved with the UAS. Why do the sonic data in b) not appear in a) with the same value? For example I don't see a data point with a flux exceeding 400 W m-2 in a) but there is one in b). If the data processing is slightly different, this is important to mention.*

We down-sampled all data to a common frequency of 5 Hz. As can be seen in Fig. 13, strong temporal gradients are present in turbulent flux. Having the center of the averaging period five minutes earlier or later can make a difference of more than 100 W m$^{-2}$. This is what happens for example at 12:35 on 22 July 2024, when the UAS flux shows a value above 400 W m$^{-2}$, but the mast values that are centered at 12:30 and 12:40 do not. In the original manuscript's Fig. 9, it is not intended to show a direct comparison of the individual points. The scatter plot for the mast shall give a baseline statistical uncertainty from the atmospheric background, it is thus beneficial to have more values including times between UAS flights. A direct comparison is shown in the time series in Fig. 11b.

51. *L256: You mention only 99m but also show 25m in the corresponding figure.*

We apologize for that inconsistency. We added the 25-m comparison later and did not update the text. We revise the manuscript to explain that sonics on both levels are compared.

52. *L259: Add information on how the filtered and unfiltered data are displayed and specify whether the error statistics correspond to filtered or unfiltered data. Apply the same filtering to the sonic data when computing error statistics and*

[Figure]

**Figure 11.** Comparison of heat flux between sonic anemometers of north and south mast (a) and between UAS and the closest mast (b). Shown are the comparisons at 25 m level (blue) and 99 m level (red). The linear regression is shown as red dotted line.

*also indicate the corresponding data points with transparent markers.*

The error statistics correspond to the filtered data. We also apply that filter now to the mast data, correlation increases, especially for the 99-m data and error is slightly reduced, see Fig. **??**

53. *L260: Correct the units.*

We correct the units, superscripting the -2.

54. *L261: Repetition: "Figure 10b shows...".*

We remove this repetitive sentence.

55. *L265-L266: In the first place, this only shows that horizontal wind speed has an impact on the uncertainty. Vertical wind speed variance may scale with wind speed and can be suggested to be the main factor behind this uncertainty, but this is not what is shown here.*

That is true, we make some implicit connections here that we did not clearly describe. In the revised manuscript we will show the vertical velocity variance with its uncertainty and how it increases with wind speed (see Fig. 12). It is evident that the temperature variance difference between sonics at the mast and UAS almost does not depend on the wind speed, while the vertical velocity does, showing some bias and increase in RMSE especially above 8 m s$^{-1}$.

56. *L269-271: Why are only UAS 15, 25, 21, 32 mentioned here when you operated so many more UAS during these flights? Are these the drones at 25m?*

Yes, these are the drones at 25 m. We hope that becomes more clear with an improved description, tables **??** and 3 and plot of the experiment (Fig. 6) as described above.

[Figure]

**Figure 12.** Dependency of the deviation of temperature variance (red) and vertical velocity variance (blue) between sonic and UAS over wind speed. Single dots show each measurement flight, the line gives the average and the error bars show the standard deviation within each velocity bin.

57. *Figure 11: What exactly is indicated by the shades and error bars? I suggest recomputing the mast data using a running average, e.g., with a 10min averaging window to provide smoother curves. The spiky data is a result of block averaging often resulting in sub-optimal start and stop times. This block averaging is likely to contribute to the second observation you highlight - substantial variability due to small offsets in time and space. I expect the comparison to become more intuitive when using running averages.*

We implement the running average with a 1-minute resolution and 10-minute window which makes the curves smoother. The shades and error bars are the determined uncertainties from the above analyses. The new plot is shown in Fig. 13.

58. *L272-282: Indicate the subfigure you refer to.*

We add references to the corresponding subfigure in the revised manuscript. The statements have to be adapted according to the newest calibration of vertical wind speed and corresponding flux calculation:

 – The SWUF-T sensors accurately capture the long-term temperature trend (Fig. 13a&e).

 – The temporal variation of temperature variance at 25 m is substantial (Fig. 13b), meaning that small offsets in time and space can lead to significant errors in direct comparisons between sonic anemometer and drone measurements. The same applies to vertical wind speed variance at 99 m (Fig. 13f).

 – The observed buoyancy flux is variable and features some distinct peaks, e.g. at 12:30 UTC at 25 m (Fig. 13d). Even more variability is found at 99 m (Fig. 13h). Some measurements by the drones are significantly higher than the sonic mast measurements, but within the range of values that is observed by the sonics during the whole period.

[Figure]

**Figure 13.** Time series of temperature $T$ (a,e), temperature variance $\sigma_T^2$ (b,f), vertical wind speed variance $\sigma_w^2$ (c,g) and sensible heat flux $H$ (d,h) for 22 July 2024. 10-minute averaged sonic anemometer measurements with a running average on south (dark blue) and north mast (light blue) are compared to drone measurements (red squares). The shades and color bars show the uncertainties as determined within this study.

59. *L276-279: Does the solar radiation or cloud cover suggest strong differential heating? The footprint of the mast and the drone at 25m are typically not strongly influenced by the surface right below. In the case studied here, winds from the west of about 8 m/s suggest a different footprint.*

We agree that the statement about footprint and an influence from the surface was very speculative and probably wrong. Estimations with improved vertical wind speed calibration do not show the effect any more. Differential heating with variable radiation and cloud cover could occur on that day, but we can unfortunately not provide any measurements of these parameters for the measurement days.

60. *L287: 5 Hz or 4 Hz as stated in L227?*

All the data is now sampled at 5 Hz. Before, the temperature sensor was internally logged at 5 Hz but interpolated along with all other data to 4 Hz. This has now been changed according to the description above.

61. *L298-290: Stationary and homogeneous flow conditions.*

    We add 'homogeneous' in the revised manuscript.

62. *L293: Specify turbulent sensible heat fluxes or buoyancy fluxes.*

    According to the description above, we change all observations to a buoyancy flux.

63. *L298: Future work should also focus on the validation of this method in a wider range of atmospheric conditions, e.g., stable conditions, very weak turbulence, and free convection.*

    That is what we mean, when we suggest to conduct measurements in a 'wider range of atmospheric conditions'. We mention the specific conditions in the revised manuscript.

64. *All subfigures should have labels if referred to as a, b, c.*

    We add the labels to all figures in the revised manuscript.

65. *The captions should include all relevant details to understand the figures or tables.*

    We modify all the captions to include all necessary details in the revised manuscript.

66. *If displaying different measurement levels or different sensors in different colours, the error statistics should also be presented separately for each category.*

    This could easily overload the plots. We believe that it strongly depends on the purpose and the message of the plot. In the cases where we believe it makes sense, we added individual statistics, but for those where the overall statistics is important, we prefer to only show one value.

67. *Labels should be consistent. I see PT100, SWUF_T, UAS.*

    We agree that there are some inconsistencies and clean that up in the revised manuscript. Where it is important to highlight the difference between slow and fast sensor, we use HYT and FWPRT, where heat fluxes are compared, not only the FWPRT is involved, but all measurements of the UAS, that is why we refer to UAS in that case. SWUF_T refers to the sensor set including fast and slow temperature sensor as well as humidity.

**References**

Baldocchi, D., Falge, E., Gu, L., Olson, R., Hollinger, D., Running, S., Anthoni, P., Bernhofer, C., Davis, K., Evans, R., Fuentes, J., Goldstein, A., Katul, G., Law, B., Lee, X., Malhi, Y., Meyers, T., Munger, W., Oechel, W., U, K. T. P., Pilegaard, K., Schmid, H. P., Valentini, R., Verma, S., Vesala, T., Wilson, K., and Wofsy, S.: FLUXNET: A New Tool to Study the Temporal and Spatial Variability of Ecosystem-Scale Carbon Dioxide, Water Vapor, and Energy Flux Densities, Bulletin of the American Meteorological Society, 82, 2415 – 2434, https://doi.org/10.1175/1520-0477(2001)082<2415:FANTTS>2.3.CO;2, 2001.

Billesbach, D. P., Arkebauer, T. J., and Sullivan, R. C.: Intercomparison of sensible and latent heat flux measurements from combined eddy covariance, energy balance, and Bowen ratio methods above a grassland prairie, Scientific Reports, 14, 21 866, https://doi.org/10.1038/s41598-024-67911-z, 2024.

Ghirardelli, M., Kral, S. T., Müller, N. C., Hann, R., Cheynet, E., and Reuder, J.: Flow Structure around a Multicopter Drone: A Computational Fluid Dynamics Analysis for Sensor Placement Considerations, Drones, 7, https://doi.org/10.3390/drones7070467, 2023.

Greene, B., Kral, S., Chilson, P., and et al.: Gradient-Based Turbulence Estimates from Multicopter Profiles in the Arctic Stable Boundary Layer, Boundary-Layer Meteorol, 183, 321–353, https://doi.org/10.1007/s10546-022-00693-x, 2022.

Kistner, J., Neuhaus, L., and Wildmann, N.: High-resolution wind speed measurements with quadcopter uncrewed aerial systems: calibration and verification in a wind tunnel with an active grid, Atmospheric Measurement Techniques, 17, 4941–4955, https://doi.org/10.5194/amt-17-4941-2024, 2024.

Liu, H., Peters, G., and Foken, T.: New Equations For Sonic Temperature Variance And Buoyancy Heat Flux With An Omnidirectional Sonic Anemometer, Boundary-Layer Meteorology, 100, 459–468, https://doi.org/10.1023/A:1019207031397, 2001.

Mauder, M. and Foken, T.: Documentation and Instruction Manual of the Eddy Covariance Software Package TK2, Arbeitsergebnisse, Universität Bayreuth, Abteilung Mikrometeorologie, ISSN 1614-8916, 46, https://doi.org/10.5194/bg-5-451-2008, 2011.

Schotanus, P., Nieuwstadt, F., and De Bruin, H.: Temperature Measurement with a Sonic Anemometer and Its Application to Heat and Moisture Fluxes, Boundary-Layer Meteorology, 26, 81–93, https://doi.org/10.1007/BF00177126, 1983.

Thielicke, W., Hübert, W., Müller, U., Eggert, M., and Wilhelm, P.: Towards accurate and practical drone-based wind measurements with an ultrasonic anemometer, Atmospheric Measurement Techniques, 14, 1303–1318, https://doi.org/10.5194/amt-14-1303-2021, 2021.

Wetz, T., Wildmann, N., and Beyrich, F.: Distributed wind measurements with multiple quadrotor unmanned aerial vehicles in the atmospheric boundary layer, Atmospheric Measurement Techniques, 14, 3795–3814, https://doi.org/10.5194/amt-14-3795-2021, 2021.

Wildmann, N. and Wetz, T.: Towards vertical wind and turbulent flux estimation with multicopter uncrewed aircraft systems, Atmospheric Measurement Techniques, 15, 5465–5477, https://doi.org/10.5194/amt-15-5465-2022, 2022.

Wildmann, N., Mauz, M., and Bange, J.: Two fast temperature sensors for probing of the atmospheric boundary layer using small remotely piloted aircraft (RPA), Atmospheric Measurement Techniques, 6, 2101–2113, https://doi.org/10.5194/amt-6-2101-2013, 2013.

---

## Author Comment (AC2)

**Towards sensible heat flux measurements with fast-response fine-wire platinum resistance thermometers on small multicopter uncrewed aerial systems.**

Norman Wildmann[1] and Laszlo Györy[1]

[1]Deutsches Zentrum für Luft- und Raumfahrt e.V., Institut für Physik der Atmosphäre, Oberpfaffenhofen, Germany
**Correspondence:** Norman Wildmann (norman.wildmann@dlr.de)

**1 Review response**

**1.1 Review General comments**

1. *This manuscript is well written and provides a topic that is relevant to the AMT readership. Fast response temperature measurements using UAS are typically achieved using fixed-wing aircraft. It is valuable to see more studies that demonstrate the ability to use rotary-wing UAS for this purpose. The authors are well qualified and the experimental techniques are suitable to demonstrate the objectives of the study: that sensible heat flux measurements can be achieved with the assistance of rotary-wing UAS. I did find it difficult at points to follow some of the data processing steps, but this can be rectified in a revision. I feel that the paper would benefit from a major revision.*

   We thank the reviewer for the objective review and good points of criticism and suggestions for improvement. We will add more detail on the processing (also in response to the second review) in order to improve the understanding and readability.

2. *The paper would benefit by having more background on eddy covariance measurements.*

   We agree that we presumed a good background knowledge of the eddy-covariance method, which is maybe not appropriate for the whole readership of the article. We add two sentences in the introduction to introduce the method and also include further references:

   "A common technique for turbulence flux measurements from stationary measurements in the ABL is the eddy-covariance method (Baldocchi et al., 2001). The eddy-covariance method directly measures the net exchange of gases, heat, and momentum between an ecosystem and the atmosphere by statistically correlating rapid fluctuations in vertical wind speed with concurrent fluctuations in the scalar of interest (e.g., gas concentration or, as in this study, temperature). [...] In order to derive accurate fluxes of sensible and latent heat in the ABL, corrections are necessary which are described in detail in Baldocchi et al. (2001) and Mauder and Foken (2011)."

3. *The paper references papers from previous work by the authors, which is appropriate, but the reader should not need to read those papers to follow the flow of the proposed study. More information from the previous studies (as related to the present work) should be provided as a summary.*

We did not explain previous work in much detail and agree that we should improve on this. We add one chapter in the revised manuscript in Section 2 describing the UAS system and previous results of calibration and spatial measurements. We also include a paragraph specifically about the vertical velocity estimation in Sect. 3:

**1.1.1 The SWUF-3D UAS**

The SWUF-3D UAS are commercially available racing drone frames of type Holybro QAV250. They are powered by a Pixhawk 4 Mini autopilot. Depending on the batteries which are used for the specific operation, the QAV250 can reach flight times up to 25 minutes. In this study, only batteries with a lower capacitiy were available, so that maximum flight times were 15 minutes and therefore, the hover periods were set to a maximum of 12 minutes. Further characteristics of the UAS are described in Wetz et al. (2021). In a fleet configuration, a multitude of drones can fly pre-defined routes synchronously and automatically. Up to twenty drones were operated during the FESSTVaL campaign (Hohenegger et al., 2023). At the research wind park WiValdi, ten drones were operated simultaneously in multiple campaigns before Wildmann and Kistner (2024, 2025). Through field tests (Wetz et al., 2021) and wind tunnel calibration (Kistner et al., 2024), the accuracy of wind speed measurement was found to be well below 0.5 m s$^{-1}$ and mostly below 0.3 m s$^{-1}$. The fleet of drones was deployed in the past to investigate spatial correlation and coherence in the ABL (Wetz et al., 2023) as well as wind speed deficit, turbulence and distinct vortices in wind turbine wakes (Wetz and Wildmann, 2023; Wildmann and Kistner, 2024, 2025).

**1.1.2 Vertical velocity esimate**

Attempting to calculate fluxes with the eddy covariance method requires synchronous measurements of temperature with vertical flow velocity $w$. As described in Wildmann and Wetz (2022), vertical velocity can be estimated from thrust and lift of the UAS. The force acting on the drone in $z$-direction $F_z$ in the body frame of the drone is a combination of gravitational force ($mg$ rotated into body frame with roll $\varphi$ and pitch $\theta$ angle), vertical accelerational forces $m\ddot{z}$, thrust $T$ and lift force $F_L$ which depends on the drag force $F_x$:

$$F_z = -mg\cos(\theta)\cos(\varphi) + m\ddot{z} + T + F_L(F_x) \tag{1}$$

Vertical velocity is derived based on a calibrated curve using the equations

$$w_b = \begin{cases} c_{z\uparrow}F_z^{b_{z\uparrow}} & F_z \geq 0 \\ c_{z\downarrow}F_z^{b_{z\downarrow}} & F_z < 0 \end{cases}. \tag{2}$$

50  Since the study in 2022, new rotors and new batteries were installed on the SWUF-3D UAS, which changes the thrust and lift behaviour. Parameters were adjusted accordingly. It also showed that it is beneficial to first rotate the forces into the geodetic coordinate system and do the vertical velocity calibration in this frame of reference, so that:

$$
w \;=\; \begin{cases} c_{z\uparrow} F_z^{g_z\uparrow} & F_z \geq 0 \\ c_{z\downarrow} F_z^{g_z\downarrow} & F_z < 0 \end{cases} . \tag{3}
$$

4. *This is related to the previous point but more information on the actual UAS would be useful.*

55  See our response above.

5. *The authors should provide more information on the sensitivity of the wind vane mode to wind speed.*

The weather vane mode is working if a roll angle larger than $1°$ can be inflicted, which is the case for wind speeds larger than $\approx 2$ m s$^{-1}$ in spanwise direction. The wind algorithm for the SWUF-3D takes spanwise wind into consideration if the UAS is not perfectly aligned. In Kistner et al. (2024) it is described how much the uncertainties increase for sideslip

60  angles up to $30°$.

6. *Are the FWPRT sensors available commercially? How prone are they to damage? Is this a limiting factor?*

The FWPRT sensors as in the SWUF-T sensor are not commercially available, but are self-designed and manufactured. Within the housing, the fine wire is not very susceptible to damage. However, hard landings or objects that will hit the wire directly can certainly damage the sensor, however that is also the case for other sensors.

65  7. *Maybe I missed it, but it is not clear to me how the vertical wind data (from the towers?) are paired with the temperature measurements from the copters?*

This may be a misunderstanding. The vertical wind data is also measured by the drones according to Wildmann and Wetz (2022). We add a section in the revised manuscript to further describe the vertical wind estimation (see above).

8. *I did not see information on the sampling rate of the sonic anemometer.*

70  The sonic anemometers have a sampling frequency of 10 Hz. In the revised manuscript they are downsampled to a rate of 5 Hz to match the UAS. This information will be explicitly given in the revised manuscript.

9. *Which instrument was used to measure humidity on the UAS?.*

For humidity, the HYT.R411 is used (HYT271 in previous versions). This is a capacitive humidity sensor.

10. *To me it seems that demonstration of the FWPRT measurements against the tower and the onboard solid state thermome-*

75  *ter would be sufficient for a study. The inclusion of measurements of flux adds extra layers of complication, which are not necessarily adequately resolved in the paper.*

It is right that temperature sensing itself is a challenging task and we want to put the focus on this. On the other hand it is

important for us demonstrate the purpose of turbulence and flux measurements very clearly, because the high temporal resolution is particularly important for these applications. We hope that the responses that are provided to the second reviewer will adequately resolve the complications, or at least transparently explain the state of the art and limitations at the current state of development.

**References**

Baldocchi, D., Falge, E., Gu, L., Olson, R., Hollinger, D., Running, S., Anthoni, P., Bernhofer, C., Davis, K., Evans, R., Fuentes, J., Goldstein, A., Katul, G., Law, B., Lee, X., Malhi, Y., Meyers, T., Munger, W., Oechel, W., U, K. T. P., Pilegaard, K., Schmid, H. P., Valentini, R., Verma, S., Vesala, T., Wilson, K., and Wofsy, S.: FLUXNET: A New Tool to Study the Temporal and Spatial Variability of Ecosystem-Scale Carbon Dioxide, Water Vapor, and Energy Flux Densities, Bulletin of the American Meteorological Society, 82, 2415 – 2434, https://doi.org/10.1175/1520-0477(2001)082<2415:FANTTS>2.3.CO;2, 2001.

Hohenegger, C., Ament, F., Beyrich, F., Löhnert, U., Rust, H., Bange, J., Böck, T., Böttcher, C., Boventer, J., Burgemeister, F., Clemens, M., Detring, C., Detring, I., Dewani, N., Duran, I. B., Fiedler, S., Göber, M., van Heerwaarden, C., Heusinkveld, B., Kirsch, B., Klocke, D., Knist, C., Lange, I., Lauermann, F., Lehmann, V., Lehmke, J., Leinweber, R., Lundgren, K., Masbou, M., Mauder, M., Mol, W., Nevermann, H., Nomokonova, T., Päschke, E., Platis, A., Reichardt, J., Rochette, L., Sakradzija, M., Schlemmer, L., Schmidli, J., Shokri, N., Sobottke, V., Speidel, J., Steinheuer, J., Turner, D. D., Vogelmann, H., Wedemeyer, C., Weide-Luiz, E., Wiesner, S., Wildmann, N., Wolz, K., and Wetz, T.: FESSTVaL: The Field Experiment on Submesoscale Spatio-Temporal Variability in Lindenberg, Bulletin of the American Meteorological Society, 104, E1875 – E1892, https://doi.org/10.1175/BAMS-D-21-0330.1, 2023.

Kistner, J., Neuhaus, L., and Wildmann, N.: High-resolution wind speed measurements with quadcopter uncrewed aerial systems: calibration and verification in a wind tunnel with an active grid, Atmospheric Measurement Techniques, 17, 4941–4955, https://doi.org/10.5194/amt-17-4941-2024, 2024.

Mauder, M. and Foken, T.: Documentation and Instruction Manual of the Eddy Covariance Software Package TK2, Arbeitsergebnisse, Universität Bayreuth, Abteilung Mikrometeorologie, ISSN 1614-8916, 46, https://doi.org/10.5194/bg-5-451-2008, 2011.

Wetz, T. and Wildmann, N.: Multi-point in situ measurements of turbulent flow in a wind turbine wake and inflow with a fleet of uncrewed aerial systems, Wind Energy Science, 8, 515–534, https://doi.org/10.5194/wes-8-515-2023, 2023.

Wetz, T., Wildmann, N., and Beyrich, F.: Distributed wind measurements with multiple quadrotor unmanned aerial vehicles in the atmospheric boundary layer, Atmospheric Measurement Techniques, 14, 3795–3814, https://doi.org/10.5194/amt-14-3795-2021, 2021.

Wetz, T., Zink, J., Bange, J., and Wildmann, N.: Analyses of Spatial Correlation and Coherence in ABL Flow with a Fleet of UAS, Boundary-Layer Meteorology, https://doi.org/10.1007/s10546-023-00791-4, 2023.

Wildmann, N. and Kistner, J.: An evaluation of different measurement strategies to measure wind turbine near wake flow with small multi-copter UAS, Journal of Physics: Conference Series, 2767, 042 004, https://doi.org/10.1088/1742-6596/2767/4/042004, 2024.

Wildmann, N. and Kistner, J.: In situ measurements of near wake dynamics with a fleet of multicopter drones, Journal of Physics: Conference Series, 3016, 012 011, https://doi.org/10.1088/1742-6596/3016/1/012011, 2025.

Wildmann, N. and Wetz, T.: Towards vertical wind and turbulent flux estimation with multicopter uncrewed aircraft systems, Atmospheric Measurement Techniques, 15, 5465–5477, https://doi.org/10.5194/amt-15-5465-2022, 2022.

---

## Author Response (AR2)

**Towards sensible heat flux measurements with fast-response fine-wire platinum resistance thermometers on small multicopter uncrewed aerial systems.**

Norman Wildmann1 and Laszlo Györy1

<sup>1Deutsches Zentrum für Luft- und Raumfahrt e.V., Institut für Physik der Atmosphäre, Oberpfaffenhofen, Germany **Correspondence:** Norman Wildmann (norman.wildmann@dlr.de)

**1 Review response**

We thank the reviewer again for a detailed review and helping to improve the manuscript. There seem to be a view misunderstandings after the last review and also some very relevant new aspects that we try to address in this response.

**1.1 Review General comments**

1. Although already suggested in the first reviews, the author's work has not yet been put in the right scientific context. Turbulence and heat flux measurements from rotary wing UAS are indeed a fairly new topic, and the novel approach presented in this study is very promising. However, at least one important article (Ghirardelli, et al. 2024) is not cited, although this is, to my knowledge, the first time the EC method has been applied in combination with a rotary-wing UAS and validated against mast-mounted ECs. Although, already suggested to include in my previous review. All relevant approaches for multirotor-based flux measurements deserve to be presented, and the proposed approach should be contrasted against the existing ones. Using a small UAS certainly has some advantages over larger systems, but it is likely to also have some disadvantages.

We have mentioned sonic-based measurements with drones in our manuscript and put our work into that context. We apologize that we missed this one latest publication and one of the citations was not compiled correctly in the last version. We assume that the reviewer means Ghirardelli, et al. 2025, which has only be finally published in May 2025, not Ghirardelli, et al. 2024? This is certainly the most suitable reference and we include it in the revised manuscript. Nevertheless, being able to do eddy-covariance flux estimations without carrying a sonic anemometer and thus being able to deploy larger fleets of UAS at lower cost and smaller weight is the novelty and unique feature of this work and is thus highlighted. The challenges for this approach are certainly much different from the challenges in carrying a sonic anemometer either fixed to the UAS or as a sling-load. Non-arguably, a sonic anemometer is an established instrument with well-known accuracy and uncertainty for flux measurements, whereas the approach to use the UAS avionics for wind measurements needs some more basic qualification as presented in this study. We extend our explanations in the introduction to further explain the differences of the approaches.

2. The fact that the separation distance impacts the comparability needs to be addressed more adequately. Comparing raw time series or time-averaged data from several different locations in rather non-stationary conditions is challenging. E.g. a good correlation can only be expected when the time lag between two time series is much shorter than the "non-stationarity time scale". At least it has to be discussed how this impacts the results, that even if all systems capture turbulence perfectly, the proximity of the UAVs to one of the masts is expected to yield higher agreement compared to the two masts standing roughly twice as far apart. A time lag correction based on cross-correlation or based on mean wind speed and separation in the along-wind direction would reduce this problem, although the problem of non-coherent smaller-scale turbulence would remain.

We agree that the comparison is challenging and subject to uncertainties that we were trying to address in our manuscript. A "non-stationarity time scale" is not a very well defined terminology, so we are not completely certain what the reviewer is expecting here. We performed instationarity tests according to Foken and Wichura (1996) and find that for the standard 30-minute period for turbulence calculation, the periods of investigation pass the test, showing stationary conditions for heat flux calculation (see Fig. 1). We also ran an alternative test configuration with 15-minute full-period averaging (similar to the drone flight time) and 2.5-minute sub-period length (in contrast to 5 minutes for the standard test). A few outliers outside the stationarity threshold of 30% can be found in that case. This is not surprising, since ever smaller averaging periods will fail to capture a proper turbulence statistic. From the default test we conclude that on typical averaging time scales of 5-30-minutes, second-order statistics are stationary for the observation periods.

Figure 1. Results of instationarity test after Foken and Wichura (1996). Dark red shows the results of the default test with 30-minute full period averaging and 5-minute sub-period length. The light red is a variation with 15-minute full period and 2.5-minute sub-period length. The dashed line shows the line for an instationarity index of 1 and the dotted lines show the threshold for  $\pm 30\%$ .

As we wrote in our last review response, we did calculate the time lag based on mean wind speed and made the correction to reduce the problem of separation between mast and drone accordingly. The time lag is of the order of a few seconds (5...10 s). We also showed that the difference in variance measurements is negligible (Fig.1 of last author response). Data shown in the last version of the manuscript already incorporated this correction. Thus, the main uncertainty is from non-coherent smaller scale turbulence, which can indeed not be adequately corrected for. That is why we show the comparison of the masts to determine some experimental uncertainty, which remains. We believe that we are very transparent with that, but do not have any possibility to reduce that uncertainty. We discuss this issue more in the revised manuscript. Not only the further distance between the masts is a drawback, but also that the masts are not aligned in the same direction as the drone and the mast.

**1.2 Review Specific comments**

1. I don't see why "sensible heat flux" has been changed to "heat flux". Maybe this is related to my previous comment on the difference between buoyancy vs. sensible heat flux, but the proposed method is suitable for measuring the sensible heat flux, so I suggest sticking to this.

Yes, we wanted to generalize this more, but agree that the main purpose of the fast temperature sensor is the sensible heat flux, so we change it back.

- 2. L8-9: I still disagree with the statement that this is the first time sensible heat fluxes have been measured (accurately) using multicopter UAS. My previous comment on this has only been partially addressed by including two suggested citations (i.e., Fuertes et al. 2019; Greene et al. 2020). However, Ghirardelli, et al. (2024) is not cited, although this is to my knowledge the first time the EC method has been applied in combination with a rotary-wing UAS (in this case, measuring the buoyancy flux using a full-scale sonic anemometer). As mentioned, I agree that this is the first time the sensible heat flux has been measured with the specific method presented in this manuscript or, more generally, applying the EC method from small UAS, but the current statement simply ignores the work done by Ghirardelli, et. al. (2024). Given the rather high discrepancies compared to the sonic anemometers, the accuracy of the measured fluxes should be put into better perspective since the uncertainty is still rather high (also for the experimental setup).
- As we wrote above, there was probably a misunderstanding, because we explicitly included the work by Ghirardelli 2023, but now it seems the reviewer rather wanted Ghirardelli 2025 as a reference, which admittedly makes more sense. With all due respect to that work, it is quite different from what we show in this study. While in Ghirardelli et al. (2025), the sonic anemometer remains the actual measurement instrument, we enable the drone itself with an additional fast-response sensor. This is shown for the first time. We will change the statement in the abstract, as in the introduction to explicitly state that we do this measurement without a sonic anemometer or other external wind sensors.
- 3. L11: I suggest simply stating the wind speed range. Furthermore, I consider 8 m/s as rather moderate wind speed and given that the minimum wind speed listed in Table 2 is 2.9 m/s, the term "low wind speed" doesn't describe the range of conditions very well.

On the full scale of possible wind speeds at the site and generally possible in the ABL, we believe that the statement is not completely wrong, but we change it to "low to moderate wind speeds" and give the rounded wind speed range as  $3...8 \text{ m s}^{-1}$  in the abstract.

4. L30: The new statement, "Another alternative is to put sonic anemometers on UAS? with the drawback of requiring much larger systems." should probably refer to Ghirardelli et al. (2023) and Thielicke et al. (2021). The cited articles are not visible and don't show up in the bibliography. The study Ghirardelli, et al. (2024), presenting a xUAS with a sonic anemometer in a sling-load configuration, is in this case more relevant than Ghirardelli, et al. (2023), which illustrates the feasibility of using sonic anemometers on xUAS based on CFD simulations. Furthermore, "larger systems" also have their advantages, e.g., longer flight times and the capability to carry more sophisticated sensors, to name a few highly relevant advantages of this approach.

We apologize for the confusion with the citations and the error in the representation of the citation in the revised manuscript. This should have been Ghirardelli et al. (2023) and Thielicke et al. (2021), but there was a mistake in the bibtex file. Larger systems certainly have their advantages, but not in this context. Flight times do actually not necessarily scale with size. In the meantime, we operate a drone below 500 g which flies 70 minutes and thus longer than larger drones that carry heavy instruments, even the one mentioned in Ghirardelli et al. (2025). Carrying more and heavy instruments is the big advantage, but maybe not necessary for heat flux measurements which are addressed in this manuscript.

5. L38: Start a new paragraph ok.

6. L31-L32: The statement is still kept too general and thus misleading. When considering a dedicated flow sensor, the separation distance between the sensor and the rotors, as well as the mechanical implementation, are important parameters. Furthermore, the sensor specifications e.g. size and sampling frequency, determine the smallest resolvable scales. For this reason, it would make sense to limit the statement to the specific sampling approach.

On the other hand, those parameters that you mention are very specific for putting a sonic anemometer on a drone. Those considerations are well described in manuscripts dealing with this approach. We change our sentence to: "Using the drone itself as a wind sensor, the smallest resolvable turbulence scales for multicopter UAS depend ..."

- 7. Figure 2: Please correct the caption of Figure 2 (red circles).We changed the caption with the correct description of symbols.
  - 8. Table 1: The Table has been improved a lot, and together with Table A1, it gives a clearer overview of which UAS have been used. However, I still lack a more detailed caption, allowing the reader to understand more easily what the different columns indicate. Some basic details on the source of the listed background conditions should also be provided here.
- 105 We expand the table caption accordingly:

"Measurement flights from 22-25 July 2024. t is the take-off time, z gives the flight heights of the drones, the average wind direction Psi, wind speed U, turbulence intensity TI, temperature T and relative humidity  $\varphi$  are based on inflow mast observations and cloud cover is determined by the crew in the field."

- 9. L156-L157: This is the definition of the sonic temperature so "...which is often referred to the 'sonic temperature'..." should better be changed to "..., the sonic temperature, ... "
  - Thanks, this is certainly a better way to put it.
- 10. Eq. 2: Please remove the +273.15K in the equation. Conversion from sonic to air temperature works fine when using either units (K or degC) consistently. The equation in its current form only works for T in degC and Ts in K.
  - Actually, since we did define T in degC before, we need to add the 273.15°C, because the equation requires the temperature to be in Kelvin. We should however subtract the value after the conversion in order to obtain  $T_s$  in degC as well and be consistent, so we change the equation to:

$$T_s = (T + 273.15^{\circ}C)(1.0 + 0.51q) - 273.15^{\circ}C \quad , \tag{1}$$

11. L161: Use "sonic temperature" instead of "equivalent sonic temperature".

Equivalent is not necessary here and will be removed, yes.

12. Fig 3: According to the authors' response, this figure should have been removed.

That is correct, we wanted to remove this figure and will definitely do it in the next revision. We had only removed the text, but not the figure by accident.

- 13. L169-L170: This new statement is still slightly misleading. The following should be corrected: The bouyancy flux is proportional to cov(Tv'w') cov(Ts'w') is a good approximation for the kinematic vertical flux of virtual temperature cov(Tv'w'), but requires scaling with density and specific heat capacity to yield the buoyancy flux Eq. 6 should be kept as is, but should indicate that this is the approximation applied in this study to estimate the buoyancy flux from the sonic anemometer data. Furthermore, it should be made clear that you compare the sensible heat flux from your UAS to the buoyancy flux from the sonic anemometers and provide some estimate on the difference between these two under the prevailing conditions. When would the contribution of the latent heat to the buoyancy flux become problematic?
- Thanks for catching these important details that we will correct in the revised manuscript. For the last point, we think that there is a misunderstanding, because with the last revision, we converted the temperature measurements from the drones to sonic temperature, using the humidity measurements by the HYT-sensor. Thus, we do compare buoyancy fluxes and with that minimize the errors in the comparison. Of course this introduces the uncertainties from the humidity sensors, but is better than comparing sensible heat flux with buoyancy flux, which becomes particularly problematic in conditions with a very small Bowen ratio, i.e. large contributions of latent heat flux.
  - 14. L205-L210: This section would benefit from some cross-references to the relevant equations. E.g., I assume that a 0D calibration would result in c1 in Eq 7. being a certain constant.

We added more references to Eq.7 and the coefficients in the text. A 0D calibration just applies an offset to the already calculated temperature based on Eq.7. You could also say it is a modification of  $c_0$  only.

15. Table C1: This table shows only 13 out of 25 sensors. Are the other 12 sensors 0D calibrated or not used at all? Does n.a. indicate that standard coefficients c0 and c1 are being used? I would assume that they can't be completely unknown since they are needed for the conversion from R to T. Delta T should not be given with varying precision. In the current form, a Delta T = 0 suggests that the bias of most sensors is below 0.00K, which is a bit hard to believe. Please also indicate your findings on the long-term stability (sensors 1 and 4) in section 4.4. The unit of c0 should be indicated (degC).

The table shows those sensors that were used within this study, the others are not used. n.a. means that the coefficients are unknown, documentation of the values is missing. The conversion from R to T is done on the microcontroller of the temperature sensor. DeltaT is the offset that was applied to those sensors where a significant bias was found. For those sensors with  $\Delta T$ =0, no bias correction was performed in post-processing. The laboratory calibration was found to be within the desired uncertainty. We add a statement in 4.4 about the two sensors that needed a bias correction and indicate the unit of  $c_0$  in the table.

- 16. Sect 4.5.: It is very helpful to have this section; however, in its current form, it is not clear how the different parameters are determined: the gravitation and acceleration term in Eq 8 can be easily determined from INS data, but it is not clear how T and FL are determined. In Eq9 and Eq10 it is not clear what c is and how it is determined, and why two different cases are treated depending on the sign of Fz. Are the coefficients c identical for the body frame and geodetic coordinate system? From a purely physical consideration, calculating a velocity from a balance of forces requires some integration in time. I assume that this is somehow accounted for by the coefficient c. However, this leaves me wondering about the initial conditions required for the integration and the sensitivity to sensor drift. I assume that small errors in the INS data would add up over time.
- Calculating a velocity from forces does not require an integration in time. The basic concept, as in horizontal wind estimation is a drag equation. Expanding on the wind algorithm goes well beyond of the scope of this paper and is extensively described in Wetz et al. (2021); Wetz and Wildmann (2022); Wildmann and Wetz (2022). The latter reference describes how T and  $F_L$  are determined, why two different cases are treated and how the coefficient c is determined.
- 17. Sect. 5.2: Fig 7a shows four Fig 7b shows five UAS. Please adjust the text and caption accordingly, also indicating that you now show two calibration flights. I think the correction is described in Sect 4.2 and according to this and Fig 7 you convert the UAV temperature to sonic temperature (not virtual temperature). The deviations between the sonic and UAV data, and also between individual UAV data, could also originate from errors in the HYT humidity readings.

We change the caption accordingly. It is sonic temperature and was not correctly changed in the last revision. When we calculate sonic temperature, it is true that humidity readings can lead to errors. A 5% uncertainty of the HYT humidity readings is the specified value, which at the given temperature, humidity and pressure yields an uncertainty of approximately 0.1 K

- 18. L249: "mean absolute deviations" instead of "average relative deviations" Done.
- 19. L264-L266: Can you provide an equation or a reference for the time-lag correction? I recommend to state that 20s is much longer than the stated time constant (HYT-271: <5s, HYT-R4211: <2s)

We add Eq. 2 and a statement that it is noteworthy that the observed time constant is much larger than the specified one.

$$T_{\rm corr} = T + \tau \frac{dT}{dt} \tag{2}$$

- 20. Table 3: Please use the same precision (number of digits) for all numeric results of the same parameter to allow for a proper comparison (not only in this table).
- 180 Done.

- 21. L306: should be "slightly higher" Done.
- 22. *L317: put citation in parentheses*Done.

- 185 23. L325: Figure 11 has been improved by also filtering out values not fitting the UAS criteria in the sonic-sonic comparison. However, it still looks like there are some differences. It is hard to count the unmasked data points, but it still looks like there are quite a few more in 11 a than in 11 b although n is almost equal. I also still have difficulties finding some distinctive data points in 11a e.g. for H > 400 W m-2, there are two data points in Fig. 11b but no such point in Fig 11a. If 11a should serve as a benchmark for the experiment uncertainty, I expect this to be based on the same data points as in 11b. This means the number of data points should be identical as well as the sonic heat flux values in 11b should 190 be found either on the south or north mast in 11 a. Since the conclusion from this analysis is that the UAS-based flux measurements fall within the experimental uncertainty, it is important that this uncertainty is benchmarked as correct as possible. The argumentation in the author's reply to my previous comments (reply 50.) that strong temporal variability is often observed, resulting in very different fluxes when shifting the start and end of the averaging intervals, leaves 195 me wondering whether different averaging intervals have been chosen. If it is the case that the averaging intervals are different, this should be mentioned to put the results in the right context, although I would prefer to see this corrected. Given the mentioned temporal variability (as shown in Figure 12), I would still advise correcting the time lag expected for the separation distance, and prevailing wind speed and direction or at least provide a proper discussion that some of the discrepancies between the two masts can be attributed to the roughly twice as long separation distance.
  - As we described within the last round of reviews, we believe that it is valuable to have as many data points as possible in the benchmark between the masts and thus use the whole period within which the UAS were flying. If we only use the exactly same data points as in the comparison between drone and mast, we do not get a very good statistic because only 32 10-minute periods remain, compared to 232 which we used for the whole period. Remember that we obtain the good statistics for the UAS comparison because we are operating multiple drones at the same time. Thus, a one-to-one comparison of the difference between mast and UAS and the two masts against each other is not reasonable in our opinion. Figure 2 shows the desired comparison matching the drone flight periods exactly. It shows that the RMSE is quite a bit larger and  $R^2$  much smaller than for the bigger dataset. For that smaller dataset you will find the datapoints with approximately 400 W m-2 in heat flux. Those points correspond to the short burst in heat flux at 12:35 UTC on 22 July, which can also be seen in Fig. 12 of the manuscript. Small shifts in the averaging period make a big difference in that case due to the high temporal gradient.
  - With regard to the time lag correction, we did apply it already, as described in the last review. We are sorry that we did not make this clear enough. We added a sentence in Section 5.4.2.
  - 24. L339-L343: mention that QAV15 and QAV25 are next to the southern mast. It is not clear that the right panels in Fig 12 correspond to the 99m level UAS and sonics. Please also indicate this in the text and the caption. Please also explain

**Figure 2.** Comparison of temperature variance between sonic anemometers of north and south mast (a) and heat flux between the two masts (b). Shown are the comparisons at 25 m level (blue) and 99 m level (red). The linear regression is shown as red dotted line.

**what the shaded areas and error bars are based on (also in the caption).**

We add an explicit statement that QAV15 and QAV25 are next to the southern mast, although it can be seen from Fig.2 and Table A1. The information about the height for each panel got lost in some revision and is added again in the next version. Error bars and shaded areas are based on the RMSE that was determined in the Section 5. We add the information to the caption.

**25. L354: How do you get to the value of 1 m for the turbulent scales that can be resolved? Would it be more accurate to state the temporal resolution of 2 Hz, since this is supported by your spectral analyses?**

It is mostly an order of magnitude that we can give here. At the lower end wind speed of 2.5 m/s and the current sampling rate of 5 Hz, 1 m is the resolution in space. Describing the temporal resolution is more straightforward and we change the text accordingly.

**225 26. Flights 69 and 70 (Table 3) only partially support the claim that turbulence can be measured accurately.**

We described in detail why especially flight # 69 is showing some stronger deviations. Flight #70 actually shows remarkably low errors. We are transparent in stating that low turbulence, stable boundary layer measurements are particularly challenging and not very well feasible with our method. We provide uncertainties with an unprecedentedly large dataset of comparison flights in this study and thus believe that our claim is well justified.

**230 References**

- Foken, T. and Wichura, B.: Tools for quality assessment of surface-based flux measurements, Agricultural and Forest Meteorology, 78, 83–105, 1996.
- Ghirardelli, M., Kral, S. T., Cheynet, E., and Reuder, J.: SAMURAI-S: Sonic Anemometer on a MUlti-Rotor drone for Atmospheric turbulence Investigation in a Sling load configuration, Atmospheric Measurement Techniques, 18, 2103–2124, https://doi.org/10.5194/amt-18-2103-2025, 2025.
- Wetz, T. and Wildmann, N.: Spatially distributed and simultaneous wind measurements with a fleet of small quadrotor UAS, Journal of Physics: Conference Series, 2265, 022 086, https://doi.org/10.1088/1742-6596/2265/2/022086, 2022.
- Wetz, T., Wildmann, N., and Beyrich, F.: Distributed wind measurements with multiple quadrotor unmanned aerial vehicles in the atmospheric boundary layer, Atmospheric Measurement Techniques, 14, 3795–3814, https://doi.org/10.5194/amt-14-3795-2021, 2021.
- Wildmann, N. and Wetz, T.: Towards vertical wind and turbulent flux estimation with multicopter uncrewed aircraft systems, Atmospheric Measurement Techniques, 15, 5465–5477, https://doi.org/10.5194/amt-15-5465-2022, 2022.